# *GhRabA4c* coordinates cell elongation via regulating actin filament–dependent vesicle transport

Xiaoguang Shang[1,2,3] , Yujia Duan[1], Meiyue Zhao[1,3], Lijie Zhu[1], Hanqiao Liu[1],
Qingfei He[1,3], Yujia Yu[1,3], Weixi Li[1], Muhammad Waqas Amjid[1] , Yong-Ling Ruan[4], Wangzhen Guo[1,2,3]

**Plant cell expands via a tip growth or diffuse growth mode. In plants, RabA is the largest group of Rab GTPases that regulate vesicle trafficking. The functions of RabA protein in modulating polarized expansion in tip growth cells have been demonstrated. However, whether and how RabA protein functions in diffuse growth plant cells have never been explored. Here, we addressed this question by examining the role of *GhRabA4c* in cotton fibers. *GhRabA4c* was preferentially expressed in elongating fibers with its protein localized to endoplasmic reticulum and Golgi apparatus. Over- and down-expression of *GhRabA4c* in cotton lead to longer and shorter fibers, respectively. GhRabA4c interacted with GhACT4 to promote the assembly of actin filament to facilitate vesicle transport for cell wall synthesis. Consistently, *GhRabA4c*-overexpressed fibers exhibited increased content of wall components and the transcript levels of the genes responsible for the synthesis of cell wall materials. We further identified two MYB proteins that directly regulate the transcription of *GhRabA4c*. Collectively, our data showed that GhRabA4c promotes diffused cell expansion by supporting vesicle trafficking and cell wall synthesis.**

## Introduction

Cell expansion is fundamental for growth and development. In plants, it occurs via either diffuse growth that refers to even growth along the entire cell facets as the case for most plant cells or tip growth where expansion is localized to the apical region of the hemispherical tip of specialized cell types such as growing pollen tubes and root hairs (Cosgrove, 2018). In both scenarios, cell expansion is controlled by a combination of cell wall deposition, turgor pressure, and cytoskeleton dynamics (Crowell et al, 2010).

Filamentous actin (F-actin) plays important roles in cell expansion by influencing the patterns in which cell wall materials are deposited (Smith & Oppenheimer, 2005). The application of actin-depolymerizing drugs, latrunculins, or cytochalasins arrested cell growth (Hepler et al, 2001). F-actin is believed to drive the long-range transport of Golgi-derived vesicles, ferrying the raw materials for the synthesis of plasma membranes and cell walls during growth (Dong et al, 2001). F-actin, which is polymerized from monomeric globular actins (G-actin), must maintain a proper rate of turnover to keep pace with the rate of cell growth. F- and G-actin interact with actin-binding proteins (ABPs), thereby exerting a great deal of functional regulation upon the actin cytoskeleton and thus the cell expansion (Dong et al, 2001; Bao et al, 2011). Investigating how the ABPs regulate actin filament dynamics is an important and ever growing field of research. It remains largely unknown, up to date, regarding the molecular mechanisms that regulate actin-mediated vesicular transport of cell wall components for cell growth.

Rab GTPases, members of the small GTPase family, are master regulators for intracellular membrane trafficking, essential for maintaining cellular functions, especially during cell expansion. Rab GTPases undergo cycling between GTP-bound active form and GDP-bound inactive form. The former associates with membrane-bound organelles, whereas the latter remains in the cytosol (Minamino & Ueda, 2019). In *Arabidopsis*, plants expressing the dominant inactive form of AtRabA2[a] caused severely abnormal cell expansion and disrupted the arrangement of cell files, whereas the cell development was rarely perturbed in plants expressing the GTP-bound active form (Chow et al, 2008). Rab GTPases are involved in the generation of specific vesicles from donor membrane organelles and the fusion of the vesicles to target organelles (Pereira-Leal & Seabra, 2000; Minamino & Ueda, 2019). There are 57 Rab GTPases in *Arabidopsis thaliana*, classified into eight groups (RabA-RabH) based on their similarity to animal Rab GTPases (Rutherford & Moore, 2002). The functions of different groups of Rab GTPases are proposed in *Arabidopsis* based on homology to mammalian

---

[1]State Key Laboratory of Crop Genetics and Germplasm Enhancement, Nanjing Agricultural University, Nanjing, China   [2]Collaborative Innovation Center for Modern Crop Production Co-sponsored by Province and Ministry, Nanjing Agricultural University, Nanjing, China   [3]The Sanya Institute of Nanjing Agricultural University, Nanjing, China   [4]Plant Science Division, Research School of Biology, The Australian National University, Canberra, Australia

Correspondence: moelab@njau.edu.cn; yong-ling.ruan@anu.edu.au

Rabs, with the exception of RabC (Lycett, 2008). To this end, RabF GTPases are associated with sterol endocytosis, whereas RabG is involved in internalization of materials into the vacuole (Lycett, 2008). On the other hand, RabE functions in vesicle trafficking from Golgi to plasma membrane, whereas RabD and RabB are involved in ER to Golgi transport of vesicles with RabH implicated in retrograde Golgi-to-ER vesicle trafficking (Mazel et al, 2004; Lycett, 2008). However, the role of the RabA GTPases has been the most difficult to assign because of functional redundancy among these genes.

Interestingly, the RabA group has greatly expanded in number and classifications in plants. For instance, compared with the two or three RabA members existing in yeast and mammalian genomes, there are 26 RabA members in *A. thaliana* and 15 in *Oryza sativa* (Rutherford & Moore, 2002; Minamino et al, 2018). Studies on *RabA* mutants suggest that vesicles carrying specific types of cargoes to the cell wall may be regulated by particular subtypes of RabAs (Preuss et al, 2004; de Graaf et al, 2005; Szumlanski & Nielsen, 2009). Analyses on a series of *Arabidopsis* T-DNA insertion lines showed that single-gene knockout of the *RabA1*, *RabA2*, and *RabA4* subclades affected the relative composition of pectin, cellulose, and hemicellulose within the cell wall, respectively (Lunn et al, 2013). Therefore, each plant RabA GTPase may have a highly specialized function that cannot be deduced based on closely related homologs but needs to be experimentally determined. Some RabA GTPases have been shown to play roles for polarized elongation in tip growing cells, such as root hairs and pollen tubes (Preuss et al, 2004; Szumlanski & Nielsen, 2009). It is unclear, however, whether and how RabA GTPases regulate cell elongation in diffuse growth cells.

Cotton fibers are single-celled seed trichomes and elongate mainly via diffuse growth mode (Seagull, 1990; Tiwari & Wilkins, 1995; Ruan, 2007) and are further investigated as a unique tip-biased diffused cell expansion mode (Yu et al, 2019). After initiating from the ovule epidermis before anthesis, the single fiber cell elongates to 3~5 cm in 15~20 d, depending on species. This makes cotton fiber an ideal system to study plant cell elongation (Ruan, 2007; Huang et al, 2021). In this study, we explored the roles of RabA4c GTPase for tip-biased diffused cell expansion, using cotton fiber as the experiment system. The analyses showed that the RabA4c protein was localized to the endoplasmic reticulum and Golgi apparatus, and active GTP-bound form of GhRabA4c is necessary for its *trans*-Golgi localization. Transgenic analyses revealed several novel observations on the functions of the *GhRabA4c* gene in cell elongation. We demonstrated that GhRabA4c could directly interact with ACT protein and affect the actin filament number and polymerization in elongating fiber cells. Moreover, the levels of major cell wall components, cellulose, hemicellulose, and pectin, were significantly affected in the *GhRabA4c* transgenic fiber cells. These effects led to longer or shorter fibers in *GhRabA4c* overexpressing or suppressing lines, respectively. We also discovered that two MYB transcription factors regulated the transcript levels of *GhRabA4c* via binding to different *cis*-elements in its promoter region. Together, the data advanced our understanding of the functions of RabA GTPase in plant cell expansion.

# Results

## GhRabA4c, a protein localized to the endoplasmic reticulum and *trans*-Golgi, is associated with cotton fiber elongation

We have previously shown through transcriptome analysis that a Rab GTPase (*Gh_A08G1321*) was significantly down-regulated in a short fiber mutant, *Ligon lintless-1* (*Li1*), compared with that in wild-type (WT) plants (Liang et al, 2015), indicating potential roles of this gene in cotton fiber cell elongation. To assess this, we isolated the full-length cDNA of this *Rab GTPase* gene from *Gossypium hirsutum* acc. TM-1. The amino acids of the cloned *Rab GTPase* displayed highest similarities with RabA4c in *A. thaliana* (*AtRabA4c*). Thus, we termed this gene as *GhRabA4c* (Fig 1A). It encoded a protein carrying the typical conserved domains of RabA GTPase, including the binding sites for phosphate or $Mg^{2+}$ and guanine nucleotide as well as the effector binding domain (Fig S1). The two-cysteine residues that play an important role in membrane localization and protein function (Li et al, 2014) were also conserved in GhRabA4c. Quantitative RT-PCR results showed that *GhRabA4c* transcripts were preferentially accumulated in elongating fibers but weakly in the other tissues of *G. hirsutum* acc. TM-1, including roots, leaves, and 20–25 days post-anthesis (DPA) fibers (Fig 1B).

*Li1* is a mutant exhibiting extremely short fibers (Kohel et al, 1974). Differences in fiber elongation between *Li1* mutant and WT were visible from three DPA onward, and the mature fibers of *Li1* mutant were much shorter than that of WT (Fig 1C). Consistent with the defects in fiber elongation, *GhRabA4c* was expressed at a much lower level in *Li1* than that in WT in elongating fibers (Fig 1D). In addition, the transcript levels of *GbRabA4c* in H7124, a *Gossypium barbadense* cultivar that has much longer fibers than *G. hirsutum* TM-1 (Hu et al, 2019), were significantly higher than that of *GhRabA4c* in TM-1 in 0–5-DPA ovules and elongating fibers (Fig 1E). This genotypic correlation between *GhRabA4c* transcript levels and fiber length strongly suggests that *GhRabA4c* plays a positive role in fiber elongation.

Next, we explored the subcellular localization of GhRabA4c. We generated mutants of GhRabA4c that were predicted either to preferentially bind GTP and thus to form the constitutively active variant ([Q74L], GhRabA4cCA) or to bind GDP and thus to form the dominant negative variant ([S29N], GhRabA4cDN) (de Graaf et al, 2005; Chow et al, 2008; Feraru et al, 2012). Wild type and variants of GhRabA4c were fused with GFP protein, respectively. Transient expression of the resulting fusions in *Nicotiana benthamiana* leaves revealed that punctate GFP signals were observed in the cytosol of epidermal cells transformed with GFP-GhRabA4c or GFP-GhRabA4cCA, and this fluorescence were colocalized with the signals from ST-mRFP, a *trans*-Golgi marker (Fig 1F and G). However, no punctate structure was detected in epidermal cells transformed with GFP-GhRabA4cDN (Fig 1H). Moreover, we found that green fluorescent signals from GFP-GhRabA4c, GFP-GhRabA4cCA, and GFP-GhRabA4cDN fusion proteins were all well overlapped with the red fluorescent signals from the mRFP-HDEL, an endoplasmic reticulum marker (Fig S2).

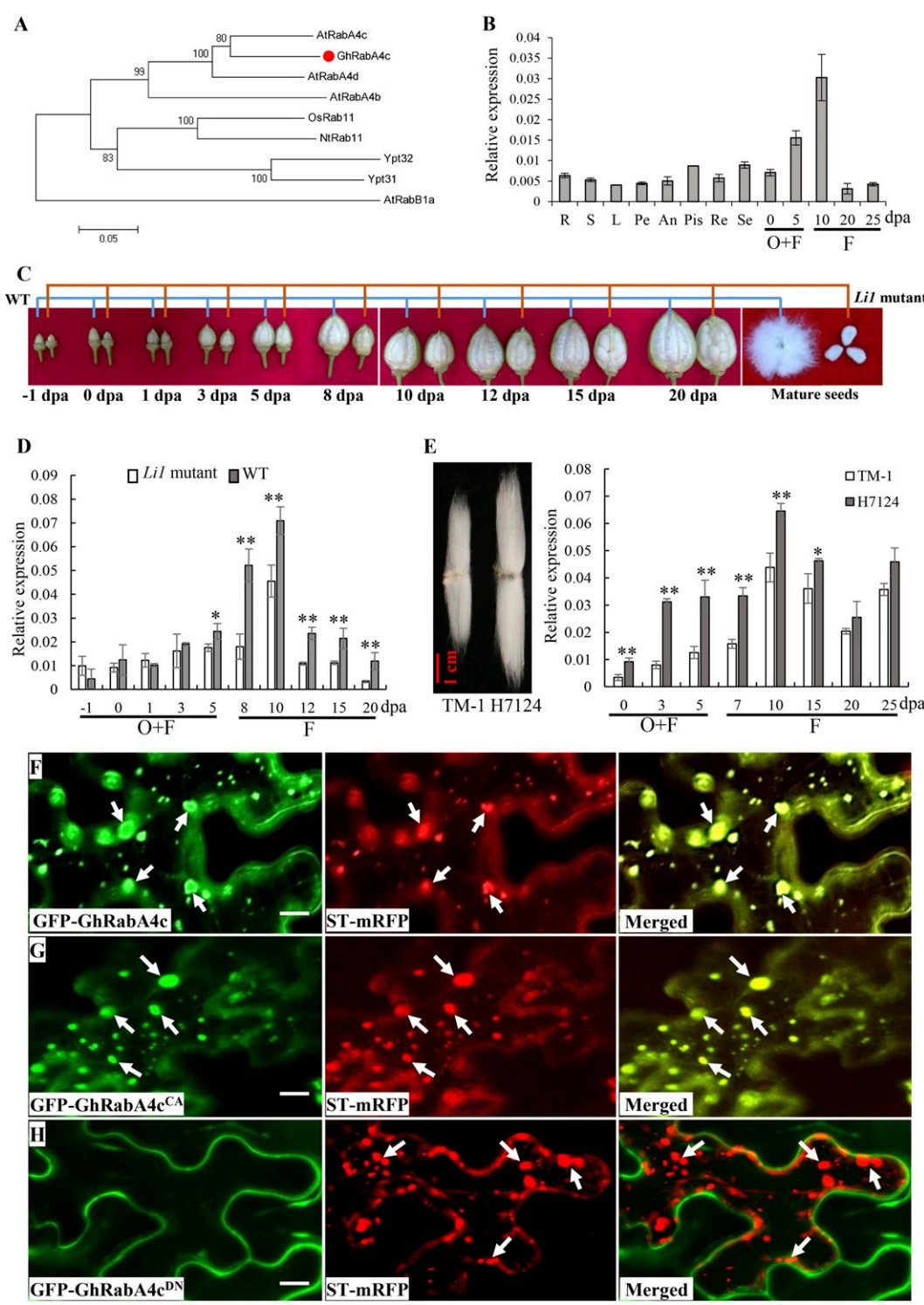

**Figure 1. Phylogenesis, expression pattern, and subcellular localization of GhRabA4c.**

**(A)** Phylogenetic analysis of GhRabA4c with homologous proteins from *Arabidopsis thaliana* (At), *Oryza sativa* (Os), *Nicotiana tabacum* (Nt), and yeast. The phylogenetic tree was inferred by using MEGA and the neighbor-joining method based on the Jones-Taylor–Thornton (JTT) model with 1,000 bootstrap replications. Percentage bootstrap scores of >50% were displayed. GhRabA4c is indicated by a red dot. GenBank accession numbers are as follows: AtRabA4b (NP_195709), AtRabA4c (NP_199607), AtRabA4d (NP_187823), OsRab11 (XP_015644018), NtRab11 (NP_001312004), Ypt31 (NP_010948), and Ypt32 (CAA96926). AtRabB1a (DQ056652) was used as a root for generating the phylogenetic tree. **(B)** Expression pattern analysis of *GhRabA4c* in tissues and organs of *G. hirsutum* TM-1. RNA was isolated from roots (R), stems (S), and leaves of 2-wk-old plants (L), petals (Pe), anthers (An), pistils (Pis), receptacle (Re), sepals (Se), 0–5-day post-anthesis ovules (O) plus attached fibers (F), and 10–25-day post-anthesis fibers. **(C)** Comparison of the phenotypes of cotton bolls and maturation seeds between WT and *Li1* mutants. Note, the smaller cotton bolls and the elongation hindered fibers on the seeds from *Li1* mutant. **(D)** Expression pattern analysis of *GhRabA4c* in ovules and elongating fibers of WT and *Li1* mutants. Note no

### GhRabA4c positively regulates cotton fiber elongation

To examine the role of GhRabA4c in cotton fiber elongation, we transformed cotton with *GhRabA4c* antisense or overexpression construct driven by the constitutive 35S promoter or RDL (*Response to Dehydration 22-Like*) fiber-specific promoter (Wang et al, 2004) (Fig S3A). Interestingly, the hypocotyls infected with *Agrobacterium* that harbors the 35S::Antisense construct became brownish, and the produced callus died during the culture process (Fig S3D). On the other hand, hypocotyls transformed with 35S::Sense, RDL::Sense or RDL::Antisense construct did generate transformed lines (Fig S3B–F).

Comprehensive PCR-based genotyping identified four 35S::Sense and three RDL::Sense or ::antisense homozygous lines at T3 generation (Fig S3G). In general, the expression levels of *GhRabA4c* were significantly higher or lower, in the overexpressed or antisense lines, respectively, relative to that in the WT plants (Fig S3H–J). No visible phenotype changes were observed between WT and transgenic plants, including plant architecture, blooming period, boll number, etc. We selected two 35S::Sense overexpression lines (lines 372 and 303, designated 35S-OE1 and 35S-OE2, respectively), two RDL::Sense overexpression lines (lines 210 and 73, designated OE1 and OE2), and two RDL::Antisense suppressed expression lines (lines 98 and 217, designated SE1 and SE2) for further analysis based on their dramatically increased or reduced *GhRabA4c* transcript levels.

Compared with WT, the mature fibers in *GhRabA4c* overexpression lines were significantly longer but were evidently reduced in the down-regulation lines (Figs 2A and B and S4A and B and Table S1). The fiber strength and micronaire value of *GhRabA4c* transgenic lines showed no significant difference with that of WT (Table S1). Scanning electron microscopy analysis did not reveal any significant differences in fiber cell density between the overexpression lines and WT plants at 0 DPA (Figs 2C and D and S4C and D). However, the number of initiated fiber cells on ovules was reduced by 55.82% in SE1 and 61.61% in SE2 compared with that of WT (Fig 2C and D), indicating *GhRabA4c* is required for proper fiber cell initiation. Consistently, compared with WT, the expression levels of *GhRabA4c* and *GhRabA4c-At* were elevated in OE1 and OE2 lines, whereas they were down-regulated in SE1 and SE2 lines, and no significant difference was detected in terms of *GhRabA4c-Dt* (Fig S5). By 1 DPA, the fiber cells of overexpressing transgenic lines and WT have properly elongated; however, some fibers appeared to remain at the initiation stage in the down-regulating lines (Fig 2C). Measurement of fiber lengths at 10, 15, and 20 DPA showed that fiber length was increased significantly in the OE lines throughout the elongation stage compared with the WT. By contrast, fiber elongation was retarded in *GhRabA4c* down-regulated lines (Figs 2E and F and

S4E–H). These findings demonstrated that *GhRabA4c* positively regulates cotton fiber elongation.

### GhRabA4c interacts with GhACTs and promotes actin filament assembly and bundling

To explore how GhRabA4c GTPase regulates the elongation of fiber cells, protein candidates potentially interacting with GhRabA4c were screened by yeast two-hybrid analysis. Screen on a yeast cDNA library made from cotton fiber mRNAs identified a protein which showed same amino acid sequence with reported GhACT4 (Li et al, 2005). Co-transformation of GAL4-BD–GhRabA4c and GAL4-AD–GhACT4 in yeast cells demonstrated that GhRabA4c interacted with GhACT4 (Figs 3A and S6). Furthermore, GFP-GhRabA4c formed filamentous structures that colocalized with actin bundles visualized by expressing ABD2-mCherry protein, an actin filament marker (Yu et al, 2019) (Fig 3B). BiFC assay showed that strong YFP fluorescence was observed in *N. benthamiana* leaf cells coexpressing GhRabA4c-nYFP with GhACT4-cYFP. By contrast, no YFP signal was detected in the negative controls that lacked RabA4c or GhACT4 (Fig 3C). Furthermore, GST pull-down assay revealed that GST-GhRabA4c did interact with GhACT4 and His-GhACT4 protein could be pulled down and detected (Fig 3D). These results demonstrated that cotton GhRabA4c indeed interacts with GhACT4 *in planta*. As *GhACT1*, a fiber specifically expressed gene, has been shown to play an important role in fiber elongation (Li et al, 2005), we further tested the interaction between GhRabA4c and GhACT1 using the yeast two-hybrid method. Interestingly, the yeast cells grew well on the triple-dropout (TDO) medium and turned blue on the quadruple-dropout (QDO) medium supplemented with X-*α*-gal (Fig S6). These results suggest that GhRabA4c not only interact with GhACT4 but also with the fiber-specific GhACT1 protein.

Next, we examined the F-actin structure in *GhRabA4c* transgenic lines by staining the actin filaments in transgenic and WT fibers with fluorescent phalloidin. Actin filaments in *GhRabA4c* overexpressing transgenic lines were significantly more bundled and denser than that in the WT, whereas the opposite was observed in *GhRabA4c* down-regulated lines (Fig 3E). Compared with WT, the occupancy, a parameter of density, of actin filament in fiber cells increased by 52.55% and 78.97% in OE1 and OE2 lines, whereas reduced by 62.85% and 46.37% in SE1 and SE2 lines, respectively (Fig 3G). Furthermore, in both transgenic and WT fibers, the actin filaments became fragmented after treatment with latrunculin B (LatB; an actin polymerization inhibitor), consistent with the occurrence of actin depolymerization. However, compared with WT, the actin cytoskeleton was still bundled to some extent in the *GhRabA4c*-overexpressed fibers but became more severely fragmented in the down-regulating fibers (Fig 3F). After LatB treatment, the

---

significant difference at the mRNA levels between WT and *Li1* mutant ovules at fiber initiation stage; however, the transcription level of *GhRabA4c* was much reduced in the *Li1* mutant as compared with that in the WT in elongating fibers. **(E)** Fiber length and expression pattern analysis of *GhRabA4c* between *G. hirsutum* TM-1 and *G. barbadense* H7124. **(F, G, H)** Subcellular localization of GhRabA4c and its constitutively active variant GhRabA4c$^{CA}$ and dominant negative variant GhRabA4c$^{DN}$. Confocal microscopy images showing that GFP-GhRabA4c and GFP-GhRabA4c$^{CA}$ were colocalized with the marker protein characteristic for the *trans*-Golgi apparatus, ST-mRFP, in *N. benthamiana* leaf epidermal cells. However, no punctate GFP signal was detected in the cytosol of *N. benthamiana* leaf epidermal cells transformed with GFP-GhRabA4c$^{DN}$. Scale bar = 10 *μ*m. Arrows indicate the *trans*-Golgi apparatus. * and ** indicate a significant difference from the WT by *t* test with *P*-values of 0.05 and 0.01, respectively. Error bars indicate SD.

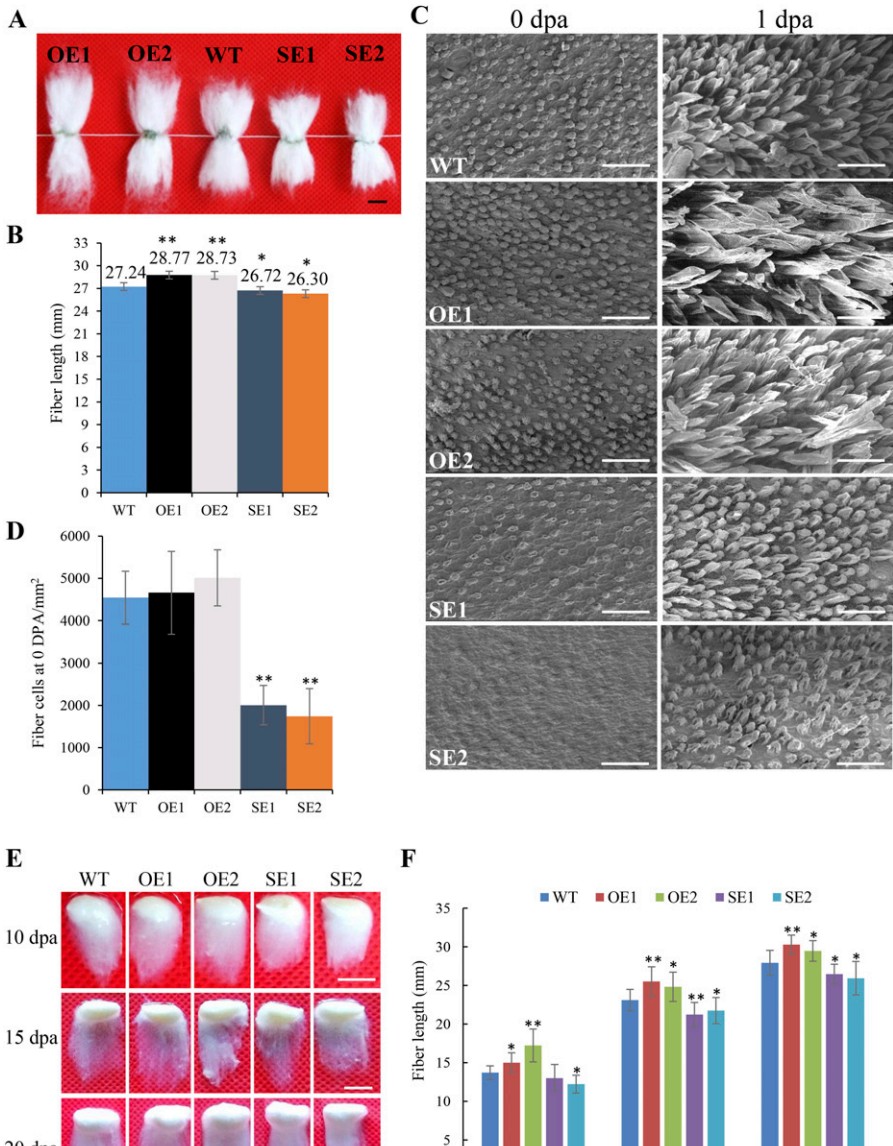

**Figure 2. *GhRabA4c* expression is positively correlated with cotton fiber elongation.**
**(A)** Phenotypic analysis of mature fiber in WT and *GhRabA4c* transgenic plants. Scale bar = 10 mm. **(B)** Mature fiber length of WT and transgenic lines. Three biological replicates of fibers from different plants (each replicate > 10 g) were used for each line. **(C)** Scanning electron micrographs of ovules at 0 and 1 days of post-anthesis (DPA) in wild-type and transgenic cotton plants. The scanning electron microscopy images were taken at a similar position in the middle of the ovules. Scale bar = 50 *μ*m. At least three ovules were observed in each sample. **(D)** Analysis of initiated fiber cell number on 0-DPA ovules in WT and transgenic plants. Six areas of equal size ($10^4$ *μm*$^2$) were investigated in each line (n = 6). **(E)** Phenotypic analysis of fibers at 10, 15, and 20 DPA in WT and transgenic plants. Scale bar = 10 mm. **(F)** Average fiber length of WT and transgenic plants at 10, 15, and 20 DPA. Fiber length was quantified and the values were averaged over nine fiber-bearing seeds of the three selected individual bolls from different plants at every stage. * and ** indicate a significant difference from the WT by *t* test with *P*-values of 0.05 and 0.01, respectively. Error bars indicate SD.

occupancy of actin filament was 34.20% and 54.09% higher in OE1 and OE2 lines but was 51.33% and 54.68% lower in SE1 and SE2 lines, respectively, compared with WT (Fig 3G). These data suggest that the actin filaments exhibited a greater stability in the fibers of OE lines but were more depolymerized in the SE lines.

### The amount of transport vesicles changed significantly in *GhRabA4c*-altered transgenic cotton fibers

To investigate whether the intracellular transportation was affected in the *GhRabA4c* transgenic cottons, the distribution of transport vesicles in elongating fiber cells was analyzed by staining with FM4-64 dye that has been shown to colocalize with

*trans*-Golgi stacks in both BY-2 cells and *Arabidopsis* root cells (Bolte et al, 2004; Berson et al, 2014). After being stained with FM4-64, middle parts of fiber cell in WT and transgenic lines were observed. Results revealed that the number and density of vesicles was significantly greater in OE1 and OE2 fibers but reduced in the SE1 and SE2 fibers as compared with that of WT (Fig 4A). More specifically, the vesicle density was increased by 35.43% and 50.73% in OE1 and OE2 lines but reduced by 36.35% and 34.55% in SE1 and SE2 lines, respectively (Fig 4B). Moreover, most of the vesicles were aggregated into patches in the SE1 and SE2 fiber cells, which may be a reason for decreased number of vesicles. These observations suggest that the amount and density of transport vesicles was affected in the *GhRabA4c* transgenic cotton fiber cells.

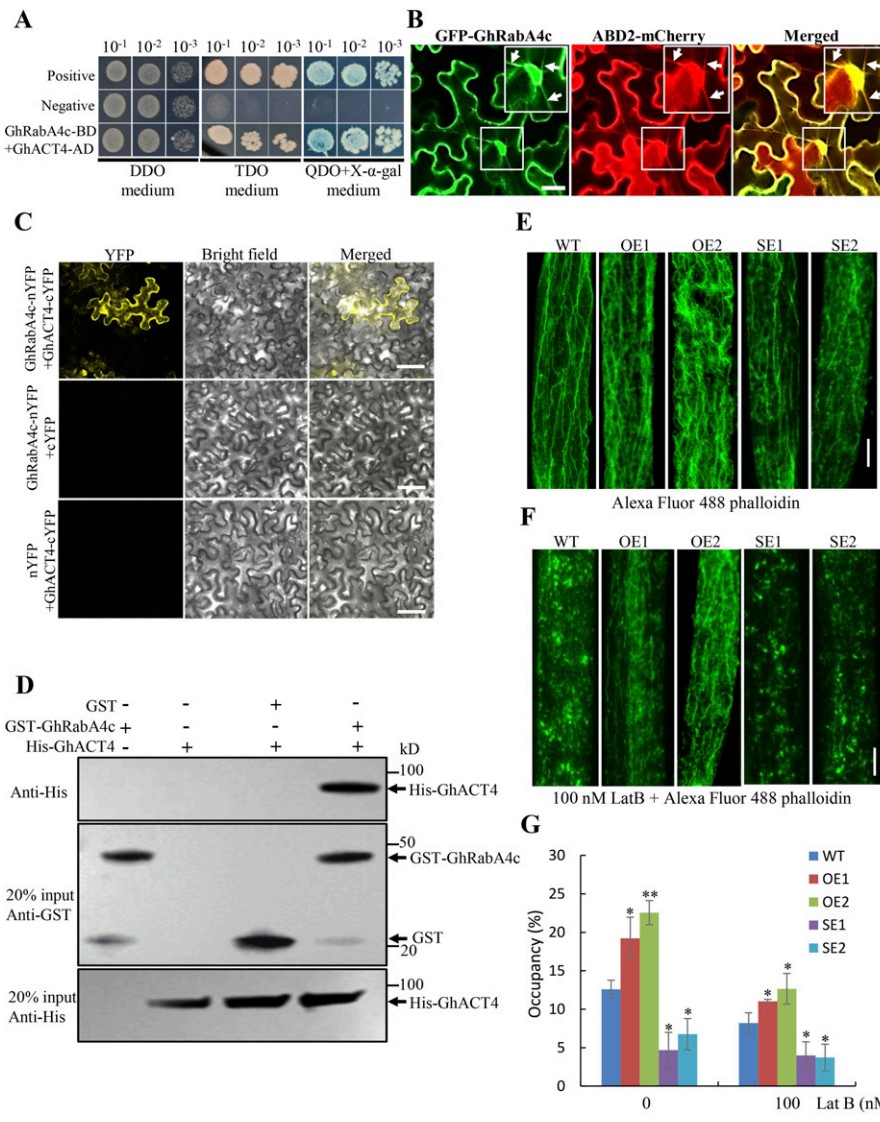

**Figure 3. GhRabA4c interacts physically with GhACT4 protein and affects actin filament assembly and bundling.**

**(A)** Yeast two-hybrid assay showing that GhRabA4c interacts with GhACT4. pGBKT7-GhRabA4c combined with pGADT7-GhACT4 conferred yeast growth on triple-dropout medium (TDO) (SD/-Leu/-Trp/-His) and quadruple-dropout medium (QDO) (SD/-Leu/-Trp/-His/-Ade+X-α-Gal) plates, respectively. pGBKT7-53 and pGBKT7-Lam were used for positive and negative controls, respectively. **(B)** Subcellular colocalization between GhRabA4c and an actin filaments marker, ABD2-mCherry, in tobacco leaf epidermal cells. Arrows indicate the actin filaments. Scale bar = 20 $\mu$m. Inset in the image is the amplified view of the boxed area. **(C)** BiFC assay showing that GhRabA4c-nYFP can interact with GhACT4-cYFP in the leaf cells of tobacco. The signals of enhanced yellow fluorescent protein (eYFP) were not detected in the corresponding controls. Scale bar = 100 $\mu$m. At least three epidermal cells were observed in each sample. **(D)** GST-GhRabA 4c protein pulls down His-GhACT4 protein in the in vitro assays. The anti-His antibody is used to detect the output protein. **(E)** F-actin organization in 10-day post-anthesis WT and *GhRabA4c* transgenic cotton fibers. Note that overexpression of *GhRabA4c* increases the density of actin filament in fibers, whereas down-regulation of *GhRabA4c* decreases the density. A maximum projection of z-slices (0.5 $\mu$m step size) was used, and three projections were analyzed in each line. Scale bar = 10 $\mu$m. **(F)** Structure of actin filaments treated by 100 nM LatB. Note that F-actin in fibers of *GhRabA4c* OE lines shows higher stability than that in fibers of WT, whereas the actin filaments are more easily depolymerized in *GhRabA4c* SE lines. A maximum projection of z-slices (0.5 $\mu$m step size) was used, and three projections were analyzed in each line. Scale bar = 10 $\mu$m. **(G)** Values of occupancy showing the index of actin alignment in WT and GhRabA4c transgenic fibers. The occupancy values indicated the proportion of the pixel numbers constituting the actin filament of the total pixel numbers constituting the fiber cell region. ImageJ software was used for measuring the pixel number, with a different threshold was applied for each image. * and ** indicate a significant difference from the WT by *t* test with *P*-values of 0.05 and 0.01, respectively. Error bars indicate SD.

## Altering *GhRabA4c* expression leading to significant changes in cell wall composition of cotton fibers

To further gain insight into the molecular pathway by which GhRabA 4c regulates fiber elongation, RNA sequencing of 10-DPA fibers from OE1 and SE1 transgenic lines was performed in comparison with that of WT. In the transcriptome analysis, an adjusted *P*-value ≤ 0.05 and an absolute value of fold change ≥ 2.0 was used to identify differentially expressed genes (DEGs) based on previous report (Zhang et al, 2015). A total of 621 DEGs were identified in the fibers of *GhRabA4c*-OE1 line, including 189 up- and 432 down-regulated genes (Fig 5A and Table S2). The up-regulated DEGs in the OE1 line were subjected to Gene Ontology (GO) analysis, which revealed the enrichment of 10 categories of cellular pathways (Fig 5B). Interestingly, three GO terms on endoplasmic reticulum network were enriched, implying the up-regulation of cell trafficking in *GhRabA4c* overexpression line. On the other hand, 491 DEGs were identified in

fibers of SE1 line in comparison with WT, including 369 up-regulated genes and 122 down-regulated genes (Fig 5C and Table S3). GO analysis from the down-regulated DEGs showed 11 significantly enriched groups in the cellular component pathway (Fig 5D). Significantly, three and four GO terms regarding intracellular lumen and cytoskeleton network were enriched, respectively, indicating the cytoskeleton and intracellular trafficking pathways were significantly affected in fibers of the SE1 line.

The expression patterns of a group of eight genes participated in cytoskeleton aggregation and polysaccharide metabolism were also revealed from the RNA-seq and verified via quantitative RT-PCR. To this end, the expression levels of *GhACT1* and *GhACT4* are significantly up- or down-regulated in the OE or SE lines, respectively (Fig 5E and F). Moreover, the expression of *GhADF5*, encoding the actin-depolymerizing factor, was down-regulated in OE lines but up-regulated in SE lines (Fig 5G). These expression data are consistent with the early finding that actin filaments were denser

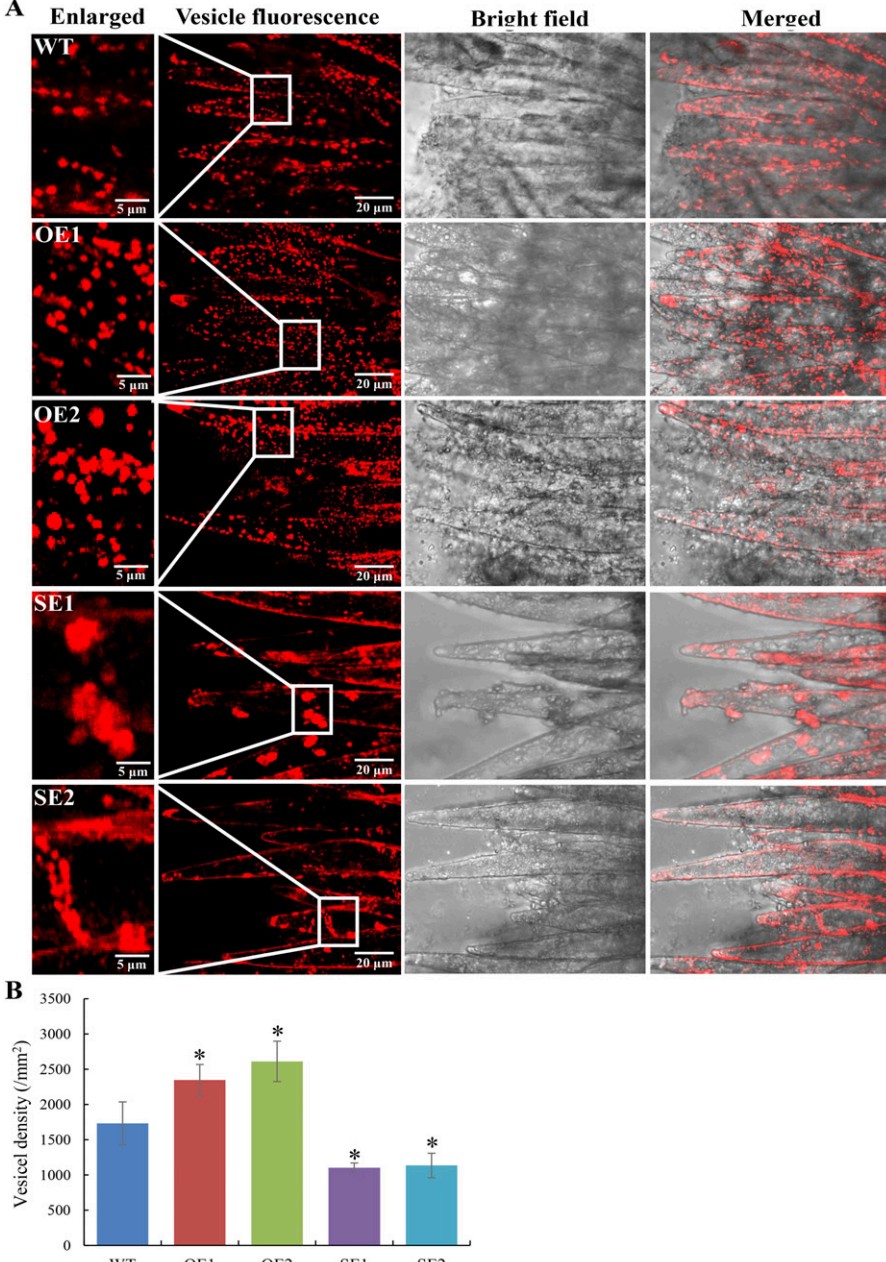

**Figure 4. Vesicle number and morphology observation in 2-day post-anthesis fiber cells.**
**(A)** Vesicles in *GhRabA4c* overexpressing lines are more than that in WT and most of the vesicles in *GhRabA4c* down-regulating lines aggregated into patches. A maximum projection of z-slices (0.5 μm step size) was used, and three projections were analyzed in each line. Photos in the first column are the enlarged views of the middle parts of fiber cells. Scale bars are indicated in the photos, respectively. **(B)** Quantification of vesicles in WT and *GhRabA4c* transgenic lines. Vesicles in each fiber cell were counted, and the fiber cell areas were measured using ImageJ software. Vesicle density was expressed as the vesicle number per square millimeter. * indicate a significant difference from the WT by *t* test with *P*-values of 0.05. Error bars indicate SD.

and more stable in the fibers of *GhRabA4c* overexpressing lines but were sparser and less stable in *GhRabA4c* down-regulating transgenic lines (Fig 3E–G). Quantitative RT-PCR analyses also revealed that genes involved in cell wall polysaccharide biosynthesis, including *GhSUS3*, *GhSUS4*, *GhUDP-GlyT1*, and *GhGlyH1*, were significantly up- or down-regulated in OE lines and SE lines, respectively (Fig 5H–K). Interestingly, *GhPL-like1*, a gene for pectin lysis, was significantly down-regulated in OE lines but was up-regulated in SE lines (Fig 5L).

Next, the levels of cellulose, hemicellulose, and pectin were measured. Compared with the WT, the cellulose content of 15-DPA fibers increased by 33.96% and 37.94% in OE1 and OE2 lines but reduced by 21.82% and 18.61% in SE1 and SE2 lines, respectively (Fig S7A). The

hemicellulose content was elevated by 10.78% and 12.51% in OE1 and OE2 lines but decreased by 12.92% and 10.58% in SE1 and SE2 lines, respectively (Fig S7B). The OE1 and OE2 fibers also displayed increases in pectin contents by 65.15% and 47.50%, whereas SE1 and SE2 lines showed 25.08% ~ 35.61% reductions in the fiber pectin contents (Fig S7C).

Further analyses revealed that the transcript levels of primary cell wall cellulose synthase genes, *GhCESA1-A*, *GhCESA1-B*, *GhCESA 3-A*, *GhCESA3-B*, *GhCESA6-B*, and *GhCESA6-C*, were dramatically up-regulated in OE lines and down-regulated in SE lines compared with WT in 10-DPA fibers (Fig S7D). Similar effects were observed for genes responsible for (i) hemicellulose biosynthesis, including those encoding cellulose synthase–like *GhCSL1*, xylosyltransferase *GhXyT1*

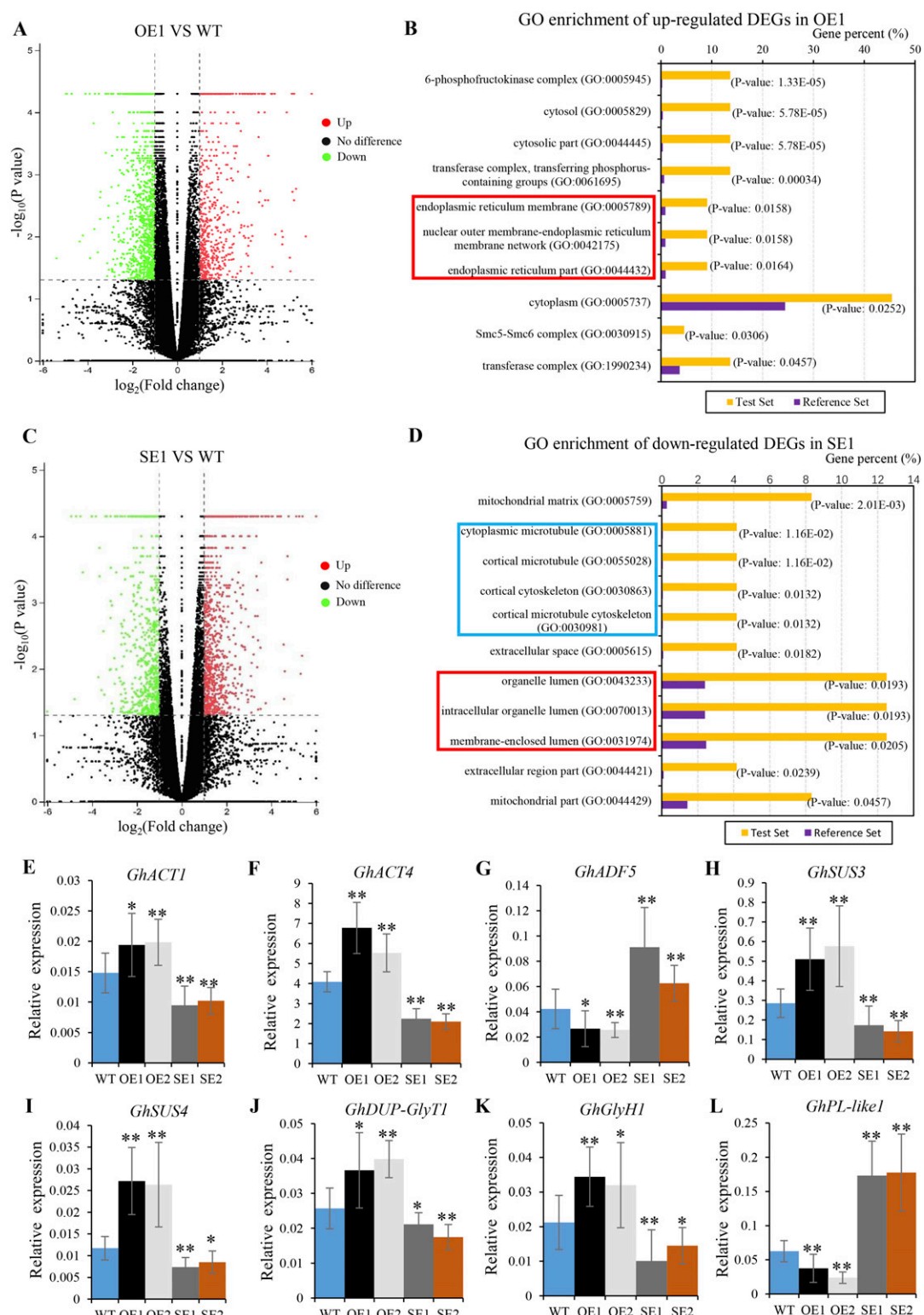

**Figure 5. RNA-seq analysis indicates *GhRabA4c* participates in endomembrane transport and affects cytoskeleton organization.**
**(A, C)** Volcano plots of differentially expressed genes (DEGs) in 10-day post-anthesis (DPA) fibers of *GhRabA4c* overexpression transgenic line OE1 (A) or *GhRabA4c* down-regulation transgenic line SE1 (C). Genes with an adjusted *P*-value ≤ 0.05 and absolute fold change value ≥ 2.0 are designated as DEGs. **(B, D)** The cellular component terms of GO analysis enriched from up-regulated DEGs in 10-DPA fibers of OE1 transgenic lines (B) or from down-regulated DEGs in 10-DPA fibers of SE1 transgenic lines (D). **(B)** In OE1 lines, GO terms regarding the endoplasmic reticulum network are significantly enriched and indicated by red box (B). **(D)** In SE1 lines, GO terms regarding cytoskeleton network and intracellular organelle lumen are significantly enriched and are indicated by blue and red box, respectively **(D)**. *P*-value of 0.05 adjusted by the

and *GhXyT2*, galactosyltransferase *GhGalT2*, and fucosyltransferase *GhFucT2* (Fig S7E), and (ii) pectin biosynthesis, including those for UDP-galactose 4-epimerase *GhUGE1*, UDP-glucuronate 4-epimerase *GhGAE6*, galacturonosyltransferase *GhGAUT10*, pectin methylesterase *GhPME2*, and pectin lyase *GhPEL1* (Fig S7F).

## GhMYB3 and GhMYB88 positively regulated the transcription of *GhRabA4c*

Given the profound impact of *GhRabA4c* on gene expression, cell wall composition, and fiber growth as described above, we attempted to explore the upstream molecular regulation on *GhRabA4c* expression. To achieve this, we first cloned the −1,546-bp promoter sequence of *GhRabA4c*, named p*GhRabA4c*. By searching PLACE and PlantCARE databases, a putative transcription start site was predicted to be located at −312 bp (the initiation codon of "ATG" was set as +1) (Fig S8A). In addition, several putative *cis*-acting elements involved in hormone responses, such as ABA, 6-BA, IAA, and GA, or transcription factor-binding including those for MYB, WRKY, and MADS Box were predicted.

To define the promoter region required to initiate *GhRabA4c* expression, the fully cloned 1,546-bp fragment (P5) and four progressively truncated fragments beginning 746 bp (P4), 546 bp (P3), 385 bp (P2), and 196 bp (P1) upstream of the initiation codon were individually fused with the GUS reporter gene (Fig S8A and B). The constructs were individually introduced into tobacco plants and screened for T3 homozygous lines. Histochemical GUS staining on tobacco plants that transformed with the whole promoter (P5 construct) showed strong GUS signals in the petal, anther, peduncle, and 10-DPA seeds (Fig S9). Very weak GUS staining was detected in 10-DPA seeds and anthers of wild-type plants and tobacco lines transformed with P1 and P2 truncated promoter constructs, respectively (Fig S8B). However, strong GUS staining was observed in the ovules and anthers of transformed with P3, P4, and P5 constructs. Consistently, quantification of GUS activity revealed significantly higher GUS activities in tobacco lines transformed with the P3, P4, and P5 constructs in comparison with that of WT and plants transformed with P1 and P2 constructs (Fig S8C). The progressively truncated p*GhRabA4c* constructs were also individually transformed into cotton fibers via particle bombardment at 0-DPA cotton ovules. It was found that strong GUS staining was observed in the ovule epidermis and attached fibers transformed with the P3, P4, and P5 construct, respectively (Fig S8D). These results suggest that the −546-bp promoter sequence is sufficient to initiate the expression of downstream gene both in tobacco and cotton fibers.

Sequence analyses identified three MYB-binding sites (MBS) that were present in P3 to P5 fragment but was absent in the P2 fragment of the p*GhRabA4c* promoter (Fig S8A). This indicates that MYB transcription factor might play an essential role in regulating the expression of *GhRabA4c* gene. At the genome level, a total of 592 MYB and MYB-like genes were identified in *G. hirsutum* acc. TM-1 (Zhang et al, 2015). Among them, 24 genes were expressed in high levels in cotton fibers (Fig S10A). Through co-expression analysis using MeV software (Howe et al, 2011), it was found that the

expression patterns of seven MYB genes were correlated to that of *GhRabA4c* (Fig S10B). Amino acid analysis indicated that these seven MYB genes have the conserved R2R3 repeats, typical characteristics of R2R3-MYBs (Fig S10C). Among these seven genes, *Gh_A05G2342* and *Gh_D05G2607* are the At- and Dt-subgenome homologous genes in tetraploid cotton; *Gh_A11G2726* and *Gh_D11G3078* are homologous to another gene; *Gh_A13G0637* and *Gh_D13G0754* are homologous genes of the third gene and the left fourth gene of *Gh_A01G1387*. Based on the sequence similarities to *Arabidopsis* MYBs and without regard to subgenome specificity, the identified cotton MYB genes were named *GhMYB3*, *GhMYB5*, *GhMYB88*, and *GhMYB4*, respectively. Quantitative PCR results show that these four MYB genes are predominantly expressed in cotton fibers, especially at the elongation developmental stage (Fig S10D).

To test whether these four MYB proteins could bind to the promoters of *GhRabA4c*, transient expression assay was conducted using tobacco leaves for the LUC reporter system. Expression vectors contain GhMYB3, GhMYB4, GhMYB5, or GhMYB88 were used as effectors, respectively (Fig 6A). The −1,546-bp promoter sequence or sequences containing the MBS1 or MBS2 motif were fused upstream of the minimal-35S::LUC to form the LUC reporter. Effectors and reporters in different combinations were co-transformed into tobacco leaves, with 35S::LUC as the positive control and effector only as the negative control (Fig 6B). The LUC activity assay indicated that GhMYB3 could bind the −1,546-bp promoter sequence and MBS2 motif, and GhMYB88 could bind the −1,546-bp promoter sequence and MBS1 motif (Fig 6B). However, GhMYB4 and GhMYB5 can bind neither the −1,546-bp promoter sequence nor the MBS1 and MBS2 elements. We further employed a dual luciferase (Dual-LUC) reporter approach to confirm these results. Expression of both GhMYB3 and GhMYB88 significantly increased the LUC/REN ratio of p*GhRabA4c*::LUC reporter (Fig 6C). Moreover, GhMYB88 significantly increased the LUC/REN ratio of MBS1::LUC reporter, and GhMYB3 significantly increased the LUC/REN ratio of MBS2::LUC reporter. The bindings of GhMYB88 on MBS1 motif and GhMYB3 on MBS2 motif were further verified using yeast one-hybrid assay (Fig 6D). Moreover, transient GUS expression analysis also confirmed GhMYB3 and GhMYB88 act as upstream regulators of *GhRabA4c* gene by regulating different MBS elements in the promoter region (Fig 6E and F). Collectively, these results demonstrate that MBS1 and MBS2 are key *cis*-elements on the *GhRabA4c* promoter for its transcription activities, and GhMYB3 and GhMYB88 are upstream *trans*-acting factors in regulating the expression of *GhRabA4c* via binding MBS2 and MBS1 site, respectively.

# Discussion

## GhRabA4c promotes cotton fiber cell elongation by enhancing the assembly and stability of actin filaments

The single-celled cotton fibers could elongate to 3.0 cm within ~16 d after anthesis. This probably makes the cotton fiber the fastest

false discovery rate. Gene percent: percentage of enriched genes comparing with background with the corresponding GO term. **(E, F, G, H, I, J, K, L)** qRT-PCR analysis of genes involved in actin filament polymerization and polysaccharide metabolism. * and ** indicate a significant difference from the WT by *t* test with *P*-values of 0.05 and 0.01, respectively. Error bars are SD of three biological replicates.

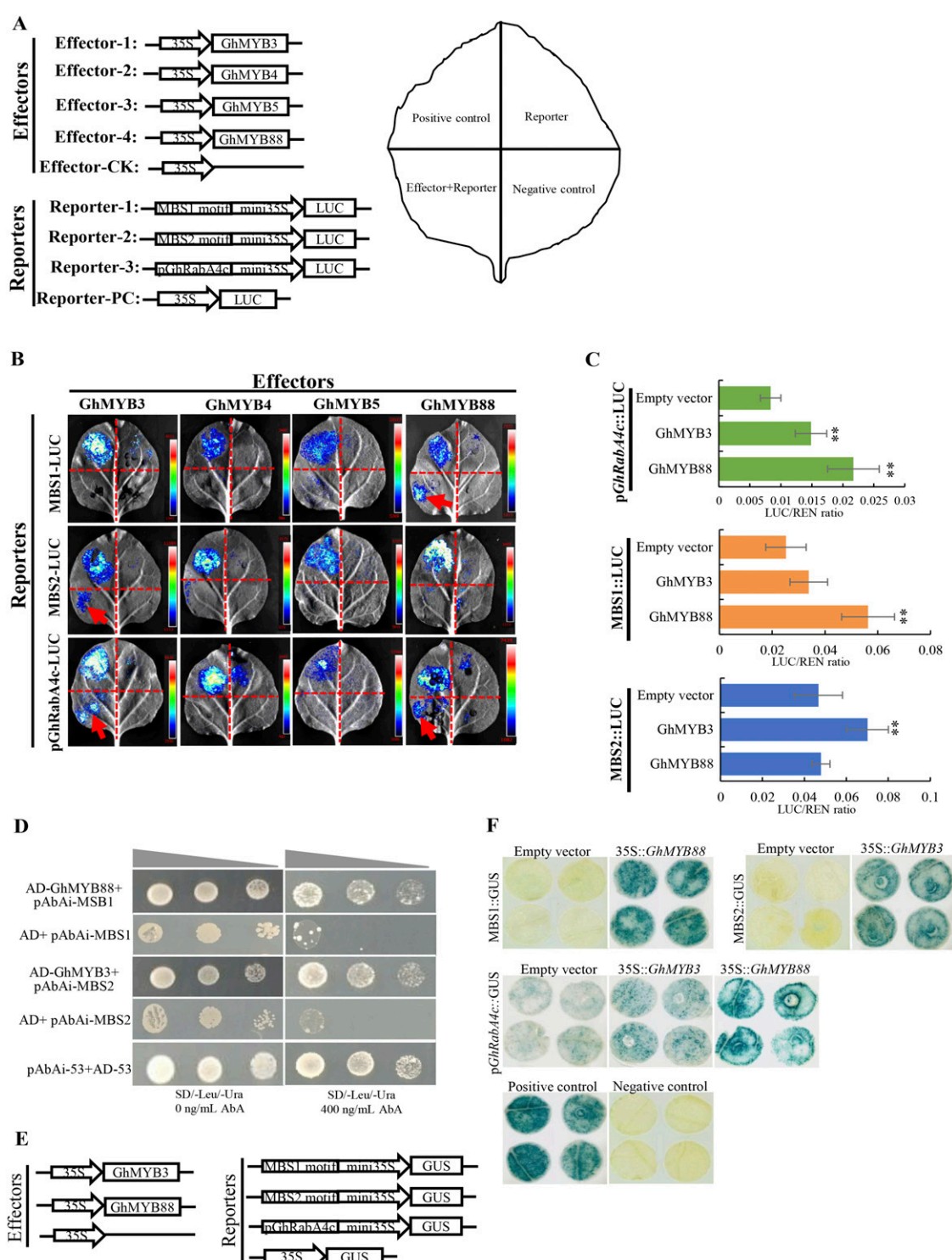

**Figure 6. GhMYB3 and GhMYB88 bind to and activate the promoters of *GhRabA4c*.**

**(A)** Schematic structures of the effector and reporter constructs and injection designing for transient expression analysis. Four candidate MYBs are inserted downstream of CaMV35S promoter to make the effectors, respectively. Six tandem repeats of the predicted MBS1 or MBS2 motif and the cloned −1,546-bp promoter sequence are fused with the minimal CaMV35S promoter (mini35S) to drive the LUC gene as reporters, respectively. The reporter construct is co-transformed with the effector of 35S:*GhMYB*s into tobacco leaves for transactivation analysis. When performing leaves injection, 35S:LUC is used as positive control and effector alone is used as negative control, respectively. **(B)** Transactivation analysis using the LUC reporter system reveals that GhMYB88 binds to MBS1 motif and GhMYB3 binds to MBS2 motif to activate the promoter of *GhRabA4c*, respectively. MBS1 motif, MBS2 motif, and −1,546-bp promoter sequence are co-transformed with the identified four candidate *GhMYB* genes, respectively. The arrows indicate the luciferase activities that are from activation of GhMYBs on the MBS motif or promoter sequence. **(C)** Transient expression assay of the promoter activity using the dual-luciferase reporter system. * and ** indicate a significant difference from the empty vector by the *t* test with *P*-values of 0.05

growing and longest single cell known in higher plants (Ruan et al, 2001). Although several genes have been shown to participate in fiber elongation, there is a lack of mechanistic insights into the regulation of this extraordinary cell growth process. In this study, we provide evidence that *GhRabA4c*, which belongs to the RabA GTPase gene family, positively controls cotton fiber length through regulating F-actin organization, vesicle transport, and the deposition of cell wall components (Fig 7).

RabA members of Rab GTPase have important functions in plant cell elongation (Minamino & Ueda, 2019). In tip-growing cells, such as root hairs and pollen tubes, certain members of the RabA GTPase proteins are involved in their polarized growth (Preuss et al, 2004; Szumlanski & Nielsen, 2009). However, the potential roles of RabA GTPase in diffuse cell growth have never been reported in plants. In this context, several lines of observations show that cotton fibers elongate mainly via diffuse growth mode (Seagull, 1990; Tiwari & Wilkins, 1995; Ruan, 2007; Yang et al, 2021 *Preprint*). In this study, we examined the role of RabA GTPase in cell growth by using cotton fiber as a model. The analyses provided several lines of evidence that *GhRabA4c* is required for proper fiber cell elongation. First, the mRNA level of *GhRabA4c* was significantly lower in elongating fibers of *Ligon Lintless-1* (*Li1*), a short fiber cotton mutant, than that its WT in *Gossypium hirstutum* (Fig 1C and D), and was much higher in *G. barbadense* cv. H7124, which harbors longer cotton fibers in comparison with that from the short fiber species, *G. hirstutum* acc. TM-1 (Fig 1E). Second, overexpressing or suppressing the expression of *GhRabA4c* resulted in longer or shorter cotton fibers, respectively (Fig 2). These results represent an unprecedented example of the function of RabA GTPase in plant cells undergoing tip-biased diffuse growth. Given that cotton is the most important textile crop worldwide, our findings provide new avenues to improve cotton fiber elongation, hence quality and yield by using genetic engineering approaches.

RabA GTPases belong to a class of vesicle transport protein. The vesicle trafficking is closely related to the dynamic of actin filament as vesicles move directionally along the F-actin network to converge or to reach the cell surface. In this study, two actin proteins, GhACT4 and GhACT1, which were highly expressed in fiber cells, were shown to interact with GhRabA4c (Figs 3A–D and S6). Various ABPs are reported to coordinate F-actin formation for cell growth, including Rab GTPase (Rasmussen et al, 2013). Studies from animal and plant cells have shown RabA group GTPase can regulate cell shape via affecting the organization of actin cytoskeleton. Down-regulation of a RabA GTPase, *Rab11*, results in disorganized actin cytoskeleton and clustering of cells in the process of tubulogenesis of *Malpighian* tubes in *Drosophila melanogaster* (Choubey & Roy, 2017). In plant cells, misregulating Rab11b activities affected actin organization in the apical and subapical regions of transformed tobacco pollen tubes (de Graaf et al, 2005). However, it remains unclear whether the changes in the dynamics and organization of the actin cytoskeleton are from a direct regulation of RabA GTPase on the actin cytoskeleton or from an indirect adjustment of the actin network to its physiological status. The demonstration here that (i) GhRabA4c interacts physically with GhACT4 and GhACT1 and (ii) F-actin structure changes in *GhRabA4c* transgenic cotton fibers underscores the direct regulation of RabA GTPase on actin filament organization (Fig 3).

### GhRabA4c is required for deposition of multiple cell wall components via regulating vesicle transport for cotton fiber elongation

Besides affecting F-actin organization, data obtained from this study also show that GhRabA4c regulates cotton fiber elongation via controlling vesicle trafficking of cell wall components. Cellulose, hemicellulose, and pectin are the major cell wall components in elongation cotton fiber cells. Cellulose is synthesized by plasma membrane localized cellulose synthase complexes which are delivered from ER to the plasma membrane via intracellular transportation. By contrast, hemicellulose and pectin are synthesized in the Golgi apparatus and packaged into tiny vesicles that fuse with the plasma membrane and thereby deliver their cargo to the wall (Cosgrove, 2005). Thus, vesicle trafficking is important for accumulation of cell wall component. In *N. benthamiana* leaf epidermal cells, punctate fluorescence signals from GFP-GhRabA4c and its active variant GFP-GhRabA4c[CA] were colocalized with the ST-mRFP–labeled *trans*-Golgi apparatus (Fig 1F and G), indicating GhRabA4c may be involved in Golgi apparatus–derived vesicle trafficking. Furthermore, staining with FM4-64, a fluorescent dye for *trans*-Golgi stacks (Bolte et al, 2004; Berson et al, 2014), revealed that the vesicles in *GhRabA4c* overexpressing lines were more than that in WT and most of the vesicles in *GhRabA4c* suppressing lines aggregated into patches (Fig 4). Previous studies showed that RabA GTPases are important for polarized cell growth via mediating specific cell wall component trafficking (de Graaf et al, 2005; Szumlanski & Nielsen, 2009; Lunn et al, 2013). As an example, *AtRabA 4d* is required for deposition of pectin in the cell wall during pollen tube polarized growth, but no content alteration of cellulose and hemicellulose was detected (Szumlanski & Nielsen, 2009). In this study, we found that expression-level changes of *GhRabA4c* led to simultaneous changes of cellulose, hemicellulose, and pectin contents in elongating cotton fibers (Fig S7A–C). These results are different from that in *Arabidopsis* and tobacco pollen tubes or root hairs where a given RabA4 controls specific cell wall component only (de Graaf et al, 2005; Szumlanski & Nielsen, 2009; Kang et al, 2011; Lunn et al, 2013). Moreover, transcriptome data indicated that the expressions of genes related to fiber development and cell wall metabolism changed significantly because of over-/down-

and 0.01, respectively. Error bars are SD of three biological replicates. **(D)** Yeast one-hybrid assay shows that GhMYB88 binds to MBS1 motif and GhMYB3 binds to MBS2 motif. 400 ng/ml aureobasidin A (AbA) is applied to suppress the basal expression of bait construct. Co-transformation of pAbAi-53 and AD-53 vectors is used for the positive control. **(E)** Constructs of effectors and reporters used for testing the binding of candidate MYB proteins on MBS motifs in transient GUS expression system. **(F)** Transient GUS expression analysis confirms the binding of GhMYB88 on MBS1 motif and GhMYB3 on MBS2 motif. GUS staining of representative leaf pieces infiltrated with reporters only or co-infiltrated with the effector and the reporter as indicated. Tobacco leaves infiltrated with 35S::GUS vector or ddH$_2$O are used as positive and negative controls, respectively.

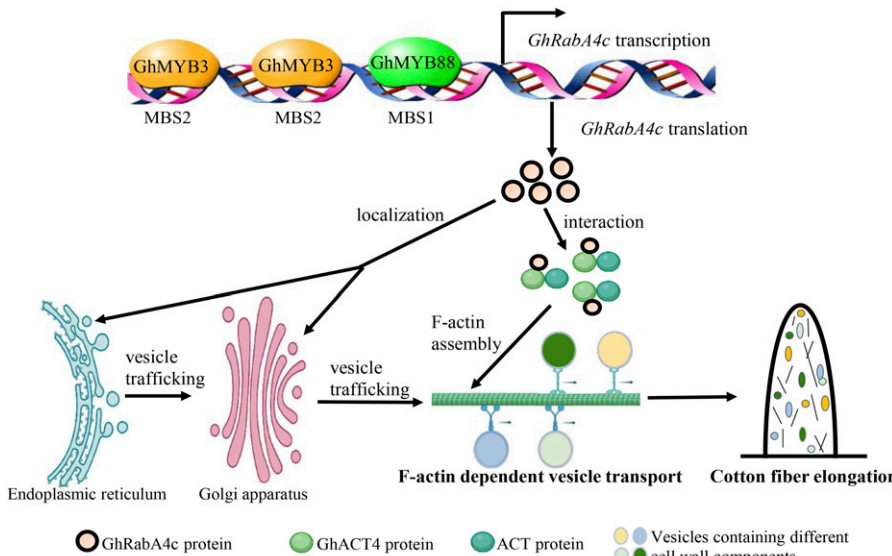

**Figure 7. A proposed model for the regulation mechanisms of *GhRabA4c* on cotton fiber elongation.**
GhMYB3 and GhMYB88 bind the *cis*-elements of MBS2 and MBS1 in the *GhRabA4c* promoter, respectively, and activate its transcription. GhRabA4c protein localizes at the endoplasmic reticulum and Golgi apparatus, and the active GTP–bound form of GhRabA4c is necessary for its *trans*-Golgi localization to function. In fiber cells, GhRabA4c promotes the assembly and bundling of actin filament via interacting with GhACT4 protein. Microfilament-dependent transport of vesicles that contain hemicellulose, pectin, or enzymes facilitating cell wall deposition is elevated in elongating cotton fibers. Finally, the fiber length is increased in *GhRabA4c* overexpression cotton plants, whereas decreased in down-regulation plants.

regulation of *GhRabA4c* (Fig 5). In *Arabidopsis*, mutation of RabA GTPases led to defect in polarized cell growth, which involved in changes of expressions of genes related to cell expansion and the metabolism of cell wall components (de Graaf et al, 2005; Szumlanski & Nielsen, 2009; Lunn et al, 2013). Similarly, the cotton fiber cell would reprogram its gene expression pattern to adapt to the changes of vesicle trafficking in the *GhRabA4c* transgenic plants. The great changes of expressions of genes as indicated in the transcriptomic analysis might be the secondary effects of metabolism adaption in fiber cell.

One possible explanation for the functional differences between GhRabA4c and *Arabidopsis* RabA GTPases in regulating cell wall component deposition is that the biological functions of RabA GTPases may have evolved to adapt the growth characteristics of cotton fibers. As described previously, cotton fibers elongate rapidly in a tip-biased diffuse growth manner (Yu et al, 2019) or diffuse growth manner (Yang et al, 2021 *Preprint*), which is different from the tip growth manner in pollen tubes and root hairs. Thus, it is likely that the function of a RabA GTPase, originally responsible for only one cell wall component for tip growth cells, has evolved to participate in the deposition of multiple cell wall components simultaneously in cotton fibers. Consistent with this notion, the C-terminal domain of GhRabA4c showed low sequence similarities with orthologous RabA GTPases in *Arabidopsis* (Fig S1). It has been shown that the C-terminal domain functions as a signal for targeting Rab GTPases to specific subcellular membranes (Li et al, 2014). In this study, GhRabA4c was localized at two types of membrane compartments, including the endoplasmic reticulum compartment and Golgi compartment (Figs 1G and S2). However, AtRabA4b was only found to be co-fractionated with a non–*trans*-Golgi membrane fraction that derived from transport vesicles in *Arabidopsis* root hairs (Preuss et al, 2004).

GhRabA4c might regulate the F-actin organization and vesicle trafficking in different forms. In vitro pull-down experiments showed that the *Escherichia coli*–expressed GhRabA4c protein, which lacks posttranslational modifications, can interact with

GhACT4 protein (Fig 3D). Thus, GhRabA4c might regulate F-actin organization in the inactive form. However, only active GhRabA4c protein co-localizes with trans-Golgi body to function in vesicle trafficking (Fig 1F–H).

## MYB transcription factors act synergistically to regulate cotton fiber elongation

Our data on promoter analysis provide additional insights into the regulatory mechanisms of *GhRabA4c* transcription. We showed that the bindings of GhMYB3 and GhMYB88 with the *cis*-elements CTG TTA and CCAACC, respectively, in the promoter of *GhRabA4c* activated its transcription (Fig 6). To the best of our knowledge, there have been no reports thus far on the identification of the upstream transcription factors of RabA GTPase in any plants.

Several MYB transcription factors have been shown to participate in cotton fiber development (Wang et al, 2021). For example, GhMML3_A12 and GhMML4_D12 are shown to regulate fuzz and lint fiber initiation, respectively (Wan et al, 2016; Wu et al, 2018). Another MYB protein, GhMYB212, functions in cotton fiber elongation through controlling the expression of a sucrose transporter gene *GhWEET12* and thus the sucrose concentration in the fiber cells (Sun et al, 2019). Interestingly, GhMYB212 directly binds to the −550-bp region of the *GhSWEET12* promoter via the *cis*-element CTGTTA that also presents in the *GhRabA4c* promoter. These results indicate that different MYB transcription factors may act synergistically to promote the elongation of cotton fiber cells. For example, GhMYB212 contributes to the regulation of sucrose transport to fiber cells, whereas GhMYB3 and GhMYB88 are responsible for the regulation of cell wall components vesicle transport inside fiber cells (Fig 6).

We noticed that the cotton hypocotyls transformed with the *GhRabA4c* antisense vector driven by the constitutive CaMV35S promoter did not survive (Fig S3D), suggesting that GhRabA4c activity is essential for cotton cell dedifferentiation to form callus. The lethal phenotype because of loss of RabA GTPase activity has not been reported previously. In *Arabidopsis*, attempts to study RabA GTPase

function by isolating insertional mutants have not revealed any observable phenotypes (Preuss et al, 2004), indicating a large degree of redundancy in the plant RabA GTPase family. In our case, the finding that constitutive down-regulation of *GhRabA4c* leads to hypocotyl death raised the possibility that potential gene redundancy previously observed in *Arabidopsis* does not apply to *GhRabA4c* in cotton.

In conclusion, our study provided novel evidence that *GhRabA4c* is essential for cotton fiber elongation, representing an example that RabA GTPase is required for expansion in tip-biased diffuse growth cells. The analyses also showed that GhRabA4c plays important role in actin filament assembly and bundling, vesicle transport, and deposition of multiple cell wall components. The identification of two MYB transcription factors modulating the transcription of *GhRabA4c* provides further insights into the molecular control of *GhRabA4c* expression (Fig 7). Collectively, the findings advanced our understanding on the roles of RabA GTPase in regulating plant cell growth and provided new perspective for genetic engineering of cotton fibers.

# Materials and Methods

### Plant materials and growth conditions

Cotton (*G. hirstutum*) plants were cultivated in the field at Wanjiang cotton experimental station, Nanjing Agricultural University (118°E, 32°N). Flowers were tagged on the day of anthesis and ovules or seeds and fibers were harvested on selected DPA. Cotton roots, stems, and leaves were collected at 14 d after germination from seedlings grown in a growth chamber (16 h: 8 h, light: dark, 28°C: 25°C). *N. benthamiana* plants were grown in a glasshouse (16 h: 8 h, light: dark, 28°C: 20°C). Fresh samples for DNA and RNA extraction and biochemical assays were frozen in liquid $N_2$ and stored at –70°C until use. Fresh fiber samples were harvested for fiber length measurement and fluorescence microscopy.

### RNA isolation and quantitative real-time PCR

Total RNA extraction and quantitative RT-PCR were performed as reported previously (Shang et al, 2020). The cotton histone 3 gene *GhHis3* (AF024716) was used as the internal control. Three biological replicates were used for each sample. Detailed primer information was shown in Table S4.

### Subcellular localization

The ORF of *GhRabA4c* was cloned into the pBINGFP4 vector to obtain the GFP-GhRabA4c construct under the control of the CaMV35S promoter. The binary vectors were transiently co-expressed in leaves of *N. benthamiana* with constructs carrying endoplasmic reticulum marker mRFP-HDEL (mRFP protein carrying endoplasmic reticulum retention signal of His-Asp-Glu-Leu; Gomord, et al, 1997) or the *trans*-Golgi apparatus marker ST-mRFP (sialyl transferase-mRFP; Saint-Jore, et al, 2002) or actin filaments marker ABD2-mCherry via the agroinfiltration method (Wang et al, 2016). Confocal

imaging was performed using a LSM780 confocal laser scanning microscope (Zeiss).

### Plasmid construction and stable transformation of cotton

The full-length ORF of *GhRabA4c* was cloned into a pBI121 vector in the sense or antisense orientation, respectively, at the *Xba*I and *Bam*HI sites downstream of the CaMV35S promoter to construct the constitutive sense and antisense plant expression vectors. In addition, sense and antisense fragments of *GhRabA4c* was cloned downstream of the RDL fiber-specific promoter to generate fiber-specific overexpression and down-regulation constructs of *GhRabA4c*, respectively (Wang et al, 2004). Binary constructs were transformed into *G. hirsutum* accession W0 by the *Agrobacterium tumefaciens* strain LBA4404 as described previously (Li et al, 2009).

### Scanning electron microscopy and fiber phenotypic analysis

Cotton ovules from similar positions on cotton plants were collected on selected DPA and processed for SEM according to Li et al (2005), using a Hitachi S-3000N scanning electron microscope. Immature fiber length was determined as described previously (Gipson & Ray, 1969). In brief, cotton seeds with fibers attached were boiled with 2.5% HCl solution for 3 min. The seeds were then transferred onto glass plates. The fibers were straightened under running water for measurement of their length. The measurement was taken at the chalazal end of the seed in all cases for consistency. Three cotton bolls from different plants were used for each measurement with three seeds each.

Mature fibers were ginned from well-developed bolls harvested simultaneously from the middle and middle-upper branches of the plants. Pools of fiber sample (each sample > 10 g) were collected to evaluate fiber quality using a high-volume instrument (HVI) (HF T9000; Premier) method. Three biological pools were used for each line.

### Yeast one-hybrid and yeast two-hybrid assay

Yeast one-hybrid (Y1H) assays were conducted with the Matchmaker Gold Yeast One-Hybrid System (Clontech). The –1,546-bp promoter fragments of *GhRabA4c* or MYB-binding sites (MBS1: CCAACC or MBS2: CTGTTA) were cloned into pAbAi vector to construct the pAbAi-bait plasmids, respectively. The plasmid was linearized, purified, and transformed into *Saccharomyces cerevisiae* Y1H Gold strain to generate a bait-reporter yeast strain. Each bait-reporter yeast strain was transformed with a pGADT7 vector (AD) and subjected to testing for the minimal inhibitory concentration of aureobasidin A (AbA). The protein–DNA interactions were tested based on the growth activity of the yeast co-transformants on SD/-Leu/-Ura medium supplemented with tested concentration of AbA according to the manufacturer's protocol. The pGADT7-53 vector was transformed to the yeast strain containing pAbAi-53 to make a positive control, and the empty vector pGADT7 was used as a negative control.

For the yeast two-hybrid (Y2H) assay of the protein–protein interaction, the full-length *GhRabA4c* was inserted into the pGBKT7 vector (bait vector), and full-length *GhACT1*/*GhACT4* was cloned into

the pGADT7 vector (prey vector) (Clontech). Plasmids were co-transferred into *S. cerevisiae* strain AH109, and then the transformants were plated on triple-dropout medium (TDO medium, SD/-Leu/-Trp/-His) (Clontech). The clones were further streaked on quadruple-dropout medium (QDO medium, SD/-Trp/-Leu/-His/-Ade) supplemented with X-α-Gal. Three independent clones for each transformation were tested. pGBKT7-53 and pGBKT7-Lam were used as positive and negative controls, respectively.

## Bimolecular fluorescence complementation (BiFC) assay

For the BiFC assays, the full-length ORF of *GhRabA4c* was inserted into p2YN-YFP, and the coding sequences of *GhACT4* were inserted into a p2YC-YFP vector. The two vectors were subsequently transformed into *A. tumefaciens* strain GV3101 and co-expressed in *N. benthamiana* leaves. YFP fluorescence signal was detected 2 d after infiltration using a LSM780 confocal laser scanning microscopy (Zeiss). The experiment was repeated for three independent biological replicates.

## In vitro GST pull-down

Interaction between GhRabA4c and GhACT4 was verified via in vitro pull-down assays. GhRabA4c was tagged with GST, and GhACT4 was tagged with histidine×6 (His). Pull-down assays were carried out by using a ProFound Pull-Down GST Protein–Protein Interaction Kit (Pierce; Thermo Fisher Scientific). The presence of His-tagged protein was detected by Western blotting using the anti-His antibody (Genscript, Inc.). GST protein from the empty pGEX4T-2 vector was used as the negative control, and His-GhACT4 fusion protein was used as the positive control.

## Fluorescent staining and microscopic analyses of the actin cytoskeleton

Observation of the actin cytoskeleton in fiber cells was performed as described previously (Wang et al, 2010). In brief, fiber-bearing seeds were carefully dissected from fresh cotton bolls at 10 DPA. F-actin in fiber cells was stained by Alexa Fluor 488 phalloidin (Invitrogen) according to the manufacturer's instructions. The seeds were washed twice with PBS (pH 7.0) for 10 min each. After fixing with 4% (wt/vol) paraformaldehyde in PBS buffer for 10 min, the fibers were cut from the seeds and washed three times with PBS and then incubated for 30 min in staining buffer (PBS solution containing 1 unit of Alexa Fluor 488 phalloidin, 40 mM Hepes [pH 7.0], 0.1% [vol/vol] Triton X-100, 3 mM dithiothreitol, 0.3 mM phenylmethylsulfonyl fluoride, and 1 mM $MgCl_2$). After briefly rinsing in PBS, the fibers were mounted onto glass slides and examined under a LSM780 confocal fluorescence microscope (Zeiss). During image acquisition, all settings, including excitation and emission wavelength (488 nm band-pass and 505–540 nm for Alexa-Fluor 488), were fixed. To assess the stability of actin filaments, fibers were treated with 100 nM latrunculin B in PBS buffer for 30 min before paraformaldehyde fixing (Yarmola et al, 2000). To evaluate actin alignment, occupancy, a statistical parameter, was employed to quantify actin density according to a previously described method (Higaki et al, 2010). In brief, the pixel numbers constituting

the actin filament and the total pixel numbers constituting the fiber cell region were measured, respectively, using ImageJ software. And the occupancy was expressed as the proportion of the former pixel numbers of the latter pixel numbers. The experiments were repeated for three independent biological replicates.

## Observation of vesicles in fiber cells

The distribution of vesicles in fiber cells was investigated according to the methods described previously (Zhao et al, 2010). Cotton bolls at two DPA were collected and the ovules inside were peeled out and incubated with 20 μg/ml FM4-64 dye dissolved in deionized water for 1 h at 25°C, darkness condition. Vesicles of fiber cells were observed under a LSM780 confocal laser microscope (Zeiss). The FM4-64 fluorescence was excited at 515 nm, with emission at 640 nm. Maximum projections of z-slices were used for further analysis. Vesicles in fiber cells were counted, and the fiber cell areas were measured using ImageJ software. Vesicle density was expressed as the vesicle number per square millimeter. At least three ovules were observed in each sample.

## RNA-seq analysis

Total RNA was isolated from 10-DPA fibers of *GhRabA4c* overexpression line OE1 and down-regulation line SE1 and wild-type samples. RNA samples were used to construct RNA-seq libraries. Sequencing was performed on an Illumina Hiseq2000 platform. After preprocessing the raw RNA-seq data with an NGS QC toolkit (Patel & Jain, 2012), all the reads were mapped to the *G. hirsutum* acc. TM-1 genome using a Hisat2 with default parameters (Kim et al, 2015; Zhang et al, 2015). Finally, the number of matched reads was determined using HTSeq and imported into R statistical software where differential expression analysis was performed using the DESeq with a cutoff of 0.05 *q* value and a fold change of >2 (Anders et al, 2015). GO analysis of the DEGs in the biological process was conducted using the R package "goseq." All samples contained three biological repeats. Expression patterns were visualized by MeV 4.7.0 (Howe et al, 2011) and clustered by the hierarchical clustering model.

## Cell wall polysaccharide content determination

The crystalline cellulose content was determined with the Upegraff method (Updegraff, 1969). The stand curve made from microcrystalline cellulose (Energy Chemical) was used for calibration. The hemicellulose content was determined with a commercial kit (Solarbio Life Science). In brief, crude cell wall of 10-DPA cotton fibers was extracted following the reported method (Zhong & Lauchli, 1993). Then, the precipitate was treated with the acid solution in the kit to convert the hemicellulose to reducing sugar, which further reacts with dinitrosalicylic acid (DNS) to produce red-brown substance. The absorption value at 540 nm reflected the hemicellulose content. Different concentrations of hemicellulose solution were used to generate the stand curve during the process. For pectin content determination, the carbazole colorimetric method was used (Zhu et al, 2017). First, the pectin fraction was extracted twice by 0.5% ammonium oxalate buffer containing 0.1%

NaBH4 (pH 4) in boiling water for 1 h each. Carbazole reagent was added into the extraction solution, and a purplish red compound was produced under a strong acidic condition. The absorption value at 530 nm was proportional to the content of galacturonic acid (GalA), which was from hydrolyzed pectin. Commercial galacturonic acid was used as a calibration standard, and the pectin content was expressed as GalA equivalents. Cotton boll from one plant was used as a biological replicate, and one experimental repeat was used as a technical replicate. All data were collected from at least three technical replicates in each of three biological replicates.

### Promoter isolation and analysis

The 5'-flanking regions of *GhRabA4c* were isolated with specific primers (Table S4). PCR products were purified, cloned into a pMD19-T vector, and sequenced (Genscript, Inc.). A 1546-bp sequence upstream of the translation initiation codon was obtained. The PLACE database (https://www.dna.affrc.go.jp/PLACE/?action=newplace) and PlantCARE program (http://bioinformatics.psb.ugent.be/webtools/plantcare/html/) were used to analyze the putative transcription start site and cis-elements.

### Transgenic tobacco generation and GUS activity determination

Five progressively truncated *GhRabA4c* promoter fragments upstream of the translation initiation codon, designated as P1 (–196), P2 (–385), P3 (–546), P4 (–746), and P5 (–1,546), were cloned, respectively. The CaMV35S promoter in the pBI121 vector was replaced by the five truncated promoter fragments via *Hind*III and *Bam*HI double digestion upstream of the GUS CDS. Furthermore, the five constructed vectors were transformed into tobacco using the *Agrobacterium*-mediated method described previously (Gallois & Marinho, 1995). Histochemical localization and activity determination of GUS was performed according to the methods described (Jefferson et al, 1987).

### Transient expression of GUS in cotton ovules

Cotton ovule culture and particle bombardment were conducted using the described methods (Kim et al, 2002). In brief, cotton bolls were collected from plants at 0 DPA. The bolls were sterilized in 75% ethyl alcohol for 90 s. Ovules were then removed from the bolls under sterile conditions and floated on solid MS medium. The ovules were cultured for 2 h in the dark at 30°C. The constructs with GUS as the reporter gene were then prepared and bombarded with a particle gun under 90 kPa. After the bombardment, the cotton ovules were immediately cultured in liquid BT medium. After 10 d of incubation, GUS staining was performed and observed with a stereoscope.

### Promoter-LUC activity assay

The coding regions of *GhMYB3*, *GhMYB4*, *GhMYB5*, and *GhMYB88* were amplified and inserted into pBI121 vector to yield the effector plasmids. The 1546-bp *GhRabA4c* promoter fragment, six repetitions of the MBS1 element or MBS2 element were ligated into the reporter

vector pGreen II 0800-LUC, respectively (Hellens et al, 2005). The effector and reporter constructs were transformed into *A. tumefaciens* GV3101 cells. *A. tumefaciens*–mediated transformation of tobacco leaf cells was conducted using the mixtures of *A. tumefaciens* cells harboring the effector and the reporter at a 1:1 ratio. Tobacco leaves infected with reporter or effector only were used as the negative control, whereas leaves infected with the pGreen-35S-LUC vector was used as the positive control. The infiltrated tobacco plants were incubated in the dark for 12 h and then were moved to normal growth conditions for 2 d at 25°C. The LUC activities were detected by histochemical method and photographed with an imaging system (Tanon 5200). Luciferase (LUC) and Renilla luciferase (REN) activities were determined via the Dual-Luciferase Reporter Assay System (Promega) with an Infinite200 Pro reader (Tecan). The transient expression assay was performed with three replicates.

## Data Availability

RNA-seq reads were submitted to the National Center for Biotechnology Information Sequence Read Archive under accession number PRJNA730924.

## Supplementary Information

## Acknowledgements

The research was supported by the Hainan Provincial Joint Project of Sanya Yazhou Bay Science and Technology City (2021JJLH0010), Key R & D Program of Jiangsu Province (BE2022382), and the Australian Research Council (DP180103834). We are grateful to Dr. Zhaosheng Kong from the State Key Laboratory of Plant Genomics, Institute of Microbiology, Chinese Academy of Sciences, for kindly providing ABD2-mCherry vector and his helpful comments. Many thanks to Dr. Peng Li from Nanjing Agricultural University for the preparation and identification of preliminary plant materials. We also thank the Bioinformatics Center of Nanjing Agricultural University for providing the data analysis platform.

### Author Contributions

X Shang: validation, investigation, and writing—original draft.
Y Duan: investigation.
M Zhao: investigation.
L Zhu: investigation.
H Liu: investigation.
Q He: investigation.
Y Yu: investigation.
W Li: investigation.
MW Amjid: investigation.
Y-L Ruan: writing—review and editing.
W Guo: conceptualization, data curation, supervision, and writing—review and editing.

**Conflict of Interest Statement**

The authors declare that they have no conflict of interest.

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
