## [Reviewer comments · Life Science Alliance]

Life Science Alliance

GhRabA4c coordinates cell elongation via regulating actin filament-dependent vesicle transport

Xiaoguang Shang, Yujia Duan, Meiyue Zhao, Lijie Zhu, Hanqiao Liu, Qingfei He, Yujia Yu, Weixi Li, Muhammad Amjid, Yong-Ling Ruan, and Wangzhen Guo

DOI: <https://doi.org/10.26508/lsa.202201450>

Corresponding author(s): Wangzhen Guo, Nanjing Agricultural University

Review Timeline:

Submission Date:	2022-03-15
Editorial Decision:	2022-04-19
Revision Received:	2022-07-16
Editorial Decision:	2022-08-15
Revision Received:	2022-08-18
Accepted:	2022-08-19

Scientific Editor: Novella Guidi

Transaction Report:

April 19, 2022

Re: Life Science Alliance manuscript #LSA-2022-01450-T

Prof. Wangzhen Guo
Nanjing Agricultural University
Weigang No1
Jiangsu 210095
China

Dear Dr. Guo,

Thank you for submitting your manuscript entitled "GhRabA4c coordinates cell elongation via regulating actin filament-dependent vesicle transport" to Life Science Alliance. The manuscript was assessed by expert reviewers, whose comments are appended to this letter. We invite you to submit a revised manuscript addressing the Reviewer comments.

Thank you for this interesting contribution to Life Science Alliance. We are looking forward to receiving your revised manuscript.

Sincerely,

B. MANUSCRIPT ORGANIZATION AND FORMATTING:

Reviewer #1 (Comments to the Authors (Required)):

Cotton fiber initiation and elongation is a process of polarization and extension of trichome cells, in which the cell cytoskeletal components and intracellular membrane reorganization are crucial for controlling this cell shape change process. In this paper, authors characterized the ER and Golgi apparatus localized Rab GTPase GhRabA4c, which is expressed at high abundance during the fiber development. Overexpression of GhRabA4c (using fiber specific RDL promoter) promoted fiber initiation and led to longer fibers, whereas its silencing reduced fiber length. In addition, GhRabA4c promotes actin filament assembly and bundling, and affects the density and number of transport vesicles in elongating fiber cells by interacting with GhACT4. Transcriptomic analysis of fiber cells revealed that expressions of genes related to cytoskeleton, intracellular trafficking and polysaccharide biosynthesis were affected, which were supported by altered cellulose, hemicellulose and pectin contents. Moreover, they characterized the promoter region of GhRabA4c and showed preliminary results that GhMYB3 and GhMYB88 are important regulator of GhRabA4c.

Major concerns:

- (1) Y2H and other experiments revealed that GhRabA4c interact with GhACT4. However, although GhACT4 expression level is high in fiber and moderate in other tissues, there is no evidence that GhACT4 plays key role in fiber initiation and elongation (Plant Cell, 17: 859-875). Authors may test the interaction of GhRabA4c with other actin proteins, especially the fiber specific GhACT1.
- (2) Is there any phenotype change in 35S::GhRabA4c plants (e.g., changed cell size and shape)?
- (3) Transcriptomic analysis revealed great changes of expressions of genes related to fiber development and the metabolism of cell wall components. As GhRabA4c is not a transcription factor, explanation of how a structural gene coordinates multiple cellular processes to change cell shape and enhance fiber elongation should be provided.
- (4) GhRabA4c is an ER and Golgi apparatus localized protein, indicating its function in intracellular transportation. Analysis of transgenic plants revealed enhanced accumulation of cell wall components such as cellulose and pectin that are outside the cell. The mechanism of how the intracellular transportation affects the cell wall structure can be discussed in more detail.
- (5) Authors show that GhMYB3 and GhMYB88 can bind to GhRabA4c promoter and activate its transcription. However, no evidence of their in vivo functions in cotton is provided. This section can be removed.

Reviewer #2 (Comments to the Authors (Required)):

The manuscript entitled "GhRabA4c coordinates cell elongation via regulating actin filament-dependent vesicle transport" by Shang et al., uncovered the Rab GTPase, GhRabA4c, plays important roles in promoting cotton fiber elongation. By testing the interaction between GhRabA4c and GhACT4 and observing F-actin networks, they found that GhRabA4c likely affects F-actin polymerization and the intracellular transportation. They also revealed that two transcription factors GhMYB3 and GhMYB88 may positively regulated the expression of GhRabA4c, resulting in cell wall composition changes in cotton fibers. This work provides important information for Rab GTPases in regulating cotton fiber development. I have no major concerns but some minor points that need to be addressed.

Comments

- As shown in Figure 2C, the initiation of fiber cells at 0 DPA was dramatically decreased in SE lines. Is the GhRabA4c also functional in regulating cotton fiber initiation? Or alternatively, are some key factors affected by the RDL::Antisense? The expression pattern of GhRabA4c SE and OE lines, as well as the transcriptional changes of its homologues in D-subgenome should also be presented.

- Line 221-222, the authors stated the actin filaments were significantly bundled in OE lines. However, quantitative data are needed to support this statement. It is also unclear whether the active form of GhRabA4c was responsible for regulating F-actin organization through the interaction with ACT4. If so, how does GhRabA4c coordinate the ACT4 combination and the trans-Golgi localization? If not, is the role of GhRabA4c regulating F-actin independent to the GTPase activity? How GhRabA4c regulates the F-actin organization by interacting with ACT4 should be further discussed.

- For better illustrating the phylogenetic relationship between Gh_A08G1321 and RabA4s from other organisms, it is recommended to use an outer group, AtRabA3s for instance, as a root when generating the phylogenetic tree.

- In figure 1C, the labels of "Li1" and "WT" was missing. Also, it is better to have the time point of GhRab4C expression pattern in Figure 1D (data at 20 DPA is missing) match the phenotypes of cotton bolls.
- The author stated that transcription levels of GbRabA4c "were significantly higher than that of GhRabA4c in TM-1 in 0 to 5 DPA ovules and elongating fibers". Hence the phenotype analysis of cotton bolls between TM-1 and H7124 were also needed.

Reviewer #3 (Comments to the Authors (Required)):

This is a manuscript that focuses on understanding the molecular machinery mediating polarized cell growth in plants, specifically cotton fiber model system. The focus of the manuscript is RabA4c small monomeric GTPase that previously has not been implicated in this process. Overall, the topic of the manuscript is very interesting since we still know relatively little about regulation of polarized cell growth in plants. Additionally, the field of Rab GTPases in plants remains quite confusing, thus, systematic analysis of the function of some of plant Rabs would be of an interest. Unfortunately, the conclusions that authors state in the manuscript are not sufficiently supported by data. For example, authors state that their study is "unprecedented example of the function of RabA GTPase in plant cells undergoing tip-biased diffuse growth", yet the studies are limited to very basic microscopy analysis and overexpression/partial knock-down of RabA4c. Consequently, most data is correlational and does not directly demonstrate that RabA4c regulates polarized cell growth. Similarly, the presumed effect of RabA4c in regulating transcription is quite confusing. Finally, there are numerous technical issues with some of the experiments (see below). All of these issues would need to be addressed before this manuscript can be accepted for publication.

- 1) Fig. 1G-H. Analyzing localization of two over-expressed proteins is always problematic since one cannot discount the possibility that these proteins co-aggregate when overexpressed. That is especially concerning since GFP-GhRabA4C localization in Figure S2 is quite different when it is co-expressed with ER marker mRFP-HDEL. The colocalization of GFP-GhRabA4c with endogenous Golgi and ER markers needs to be shown. Also, it would be good to know the level of overexpression of GFP-GhRabA4c (as compared to endogenous GhRabA4c).
- 2) Figure 2A-B. The change in fiber length is modest at best. The only more dramatic difference is at 1 DPA stage. That questions the importance of GhRabA4c for cell elongation.
- 3) Figure 3A. Y2H assays is missing some key controls. First, colonies of Double-drop out media needs to be shown to ensure that "negative" control yeast are viable. Second, pGBKT7-GhRabA4c and pGADT7-GhACT4 alone needs to be shown to control for possible auto-activation.
- 4) Figure 3B. Why GFP-GhRabA4c localization is so different compared to the ones shown in colocalization with Golgi experiments?
- 5) Figure 4. Based on bright field images, the area imaged for SE1 and SE2 is quite different. I am not even convinced these are the vesicles. How was density quantified?
- 6) Figure 5. Not quite sure about scientific rationale by doing RNAseq analyses of lines expressing different levels of GhRabA4c. If GhRabA4c regulates membrane transport and actin dynamics, why authors would expect that it will directly affect transcription?
- 7) Figure 5E-L. How authors know that changes in particular gene expression is the result of GhRabA4c levels or simple clonal variations. Do the same changes in expression observed (by qPCR) if using OE2 or SA2 lines?

Response to Review 1#

Thank you for your review. The authors had accepted and corrected the MS as you suggested. The manuscript had been re-written and revised seriously following your comments. The explanations point to point as follows:

1. Y2H and other experiments revealed that GhRabA4c interact with GhACT4. However, although GhACT4 expression level is high in fiber and moderate in other tissues, there is no evidence that GhACT4 plays key role in fiber initiation and elongation (Plant Cell, 17: 859-875). Authors may test the interaction of GhRabA4c with other actin proteins, especially the fiber specific GhACT1.

The author's answer: Thanks for your comments. It's a good suggestion to test the interaction of GhRabA4c with the fiber specific GhACT1. We employed yeast two hybrid experiment to perform the test. Interestingly, the yeast cells grew well on the triple dropout (TDO) medium and turned blue on the quadruple dropout (QDO) medium supplemented with X- α -gal. These results suggest that GhRabA4c not only interact with GhACT4, but also with the fiber specific GhACT1. We have added the interaction results between GhRabA4c and GhACT1 in the Results (Please see line 236-242 and Appendix Fig S7), and discussed the implications of the interactions between GhRabA4c and GhACT1/ GhACT4 in the Discussion (Please see line 466-468).

2. Is there any phenotype change in 35S::GhRabA4c plants (e.g., changed cell size and shape)?

The author's answer: Thanks for your question. In the 35S::GhRabA4c plants, the mature fibers of 35S-OE1 and 35S-OE2 lines are significantly longer than that of WT. Detection of immature fiber length at 10, 15, and 20 DPA also demonstrated that fibers of 35S-OE1 and 35S-OE2 lines exhibited longer fibers. We have put these results in the revised manuscript as supplemental figure S4. Please see line 183-207.

3. Transcriptomic analysis revealed great changes of expressions of genes related to fiber development and the metabolism of cell wall components. As GhRabA4c is not a transcription factor, explanation of how a structural gene coordinates multiple cellular processes to change cell shape and enhance fiber elongation should be provided.

The author's answer: Thanks for your comments. GhRabA4c is a Rab GTPase that functions in regulating vesicle trafficking in plant cell. Previous studies in *Arabidopsis* showed that RabA GTPases are important for polarized cell growth, which involved in changes of expressions of genes related to cell expansion and the metabolism of cell wall components (de Graaf et al., 2005; Szumlanski and Nielsen, 2009; Lunn et al., 2013). As you pointed out, the expressions of genes related to fiber development and cell wall metabolism changed significantly due to over-/down-regulation of *GhRabA4c*. We speculate that the fiber cell would reprogram its gene expression pattern to adapt to the changes of vesicle trafficking in the transgenic plants. The changes of expressions of genes related to fiber development and the metabolism of cell wall components might

be the secondary effects of metabolism adaption in fiber cell. We have provided these discussions in the revised manuscript. Please see line 514-526.

4. GhRabA4c is an ER and Golgi apparatus localized protein, indicating its function in intracellular transportation. Analysis of transgenic plants revealed enhanced accumulation of cell wall components such as cellulose and pectin that are outside the cell. The mechanism of how the intracellular transportation affects the cell wall structure can be discussed in more detail.

The author's answer: Thanks for your suggestion. We have added the discussion in the revised manuscript. Please see line 488-496.

5. Authors show that GhMYB3 and GhMYB88 can bind to GhRabA4c promoter and activate its transcription. However, no evidence of their in vivo functions in cotton is provided. This section can be removed.

The author's answer: Thanks for your suggestion. We have taken your suggestion seriously. Although no evidence of in vivo functions of *GhMYB3* and *GhMYB88* in cotton is provided in this manuscript, we still consider that the results that *GhMYB3* and *GhMYB88*, with the similar expression patterns with *GhRabA4c* in cotton fibers, regulate expressions of *GhRabA4c* are important for this study. Thus, we retained this section in the revised manuscript. Thanks for your suggestion again.

Response to Reviewer 2#

Thank you for reading this paper and providing kind comments. The authors had accepted and corrected the MS as you suggested. The explanations point to point as follows:

1. As shown in Figure 2C, the initiation of fiber cells at 0 DPA was dramatically decreased in SE lines. Is the GhRabA4c also functional in regulating cotton fiber initiation? Or alternatively, are some key factors affected by the RDL::Antisense? The expression pattern of GhRabA4c SE and OE lines, as well as the transcriptional changes of its homologues in D-subgenome should also be presented.

The author's answer: Thanks for your comments. Following your suggestion, we further detected the expression pattern of *GhRabA4c* (Primers that can amplify *GhRabA4c-At* and *GhRabA4c-Dt*, respectively), *GhRabA4c-At* and *GhRabA4c-Dt* in 0 DPA ovules, in which the fiber cells initiate. Please see appendix figure S5 and line 198-201 in the revised manuscript. As expected the expression levels of *GhRabA4c* and *GhRabA4c-At* was elevated in OE1 and OE2 lines, while was down-regulated in SE1 and SE2 lines, compared with WT. No significant difference was observed between transgenic fibers and WT in terms of expression levels of *GhRabA4c-Dt*. When fiber initiates, the cell wall components are synthesized and transported to the cell wall, but not as large amount as that fibers need at elongating stage. Thus, down-regulation of *GhRabA4c* will decrease the initiation of fiber cells at 0 DPA, but elevated expression level of *GhRabA4c* may not cause changes in fiber initiation.

2. Line 221-222, the authors stated the actin filaments were significantly bundled in OE lines. However, quantitative data are needed to support this statement. It is also unclear whether the active form of GhRabA4c was responsible for regulating F-actin organization through the interaction with ACT4. If so, how does GhRabA4c coordinate the ACT4 combination and the trans-Golgi localization? If not, is the role of GhRabA4c regulating F-actin independent to the GTPase activity? How GhRabA4c regulates the F-actin organization by interacting with ACT4 should be further discussed.

The author's answer: Thanks for your comments. Your suggestions that make quantitative analysis on bundled actin filaments is a good piece of advice. We have tried to measure the thickness of filaments in the fiber cells. However, due to the actin filaments are folded and curved in the fiber cells and they are not on a plane, the obtained data is not accurate. Thus, we gave up doing that. Indeed, as shown in Fig 3G, the occupancy that indicated the proportion of the pixel numbers constituting the actin filament of the total pixel numbers constituting the fiber cell region are the quantitative data of bundled actin filaments.

As you suggested, we consider that the role of GhRabA4c regulating F-actin is independent to the GTPase activity. *In vitro* pull down experiments showed that the *E. coli* expressed GhRabA4c protein, which lack of post-translational modification, can interact with ACT4 protein. Thus, GhRabA4c can regulate F-actin organization through interact with GhACT4 and GhACT1 (supplemented results showed that GhRabA4c also interacts with GhACT1) in the inactive form, and functions in vesicle trafficking via co-localizing with Golgi body in the active form. We have added this in the Discussion. Please see line 544-549.

3. For better illustrating the phylogenetic relationship between Gh_A08G1321 and RabA4s from other organisms, it is recommended to use an outer group, AtRabA3s for instance, as a root when generating the phylogenetic tree.

The author's answer: Thanks for your suggestions. We have modified the phylogenetic tree. Please see the revised Figure 1A.

4. In figure 1C, the labels of "Li1" and "WT" was missing. Also, it is better to have the time point of GhRab4C expression pattern in Figure 1D (data at 20 DPA is missing) match the phenotypes of cotton bolls.

The author's answer: Thanks for your suggestions. We have modified and added the labels of "Li1" and "WT" in Figure 1C. The expression levels of *GhRab4C* at 20 DPA in WT and *Li1* mutant have also been added in Figure 1D. Thanks again for your helpful suggestion.

5. The author stated that transcription levels of GhRabA4c "were significantly higher than that of GhRabA4c in TM-1 in 0 to 5 DPA ovules and elongating fibers". Hence the phenotype analysis of cotton bolls between TM-1 and H7124 were also needed.

The author's answer: Thanks for your suggestion. TM-1 is a standard *G. hirsutum* accession, and whereas H7124 is a *G. barbadense* accession that produces exceptionally high quality fibers. Both of these two accessions are widely used for genetic studies, especially for fiber development studies in the cotton research community. At this time, as cotton just begins to flower, it will take long time to collect samples and make phenotype analysis on the cotton bolls between TM-1 and H7124. Moreover, as lots of studies have shown the developmental differences in fibers between TM-1 and H7124 (as exemplified in the following figure), we did not provide the phenotype analysis herein but inserted the reference and revised the text in the manuscript. Please see line 140. Thanks for your suggestion again.

Phenotypes of fiber-bearing seeds in Hai7124 and TM-1 plants (Hu *et al.*, 2019).

Response to Reviewer 3#

Thank you for your time and constructive comments. The authors had accepted and corrected the MS as you suggested. The explanations point to point as follows:

1. Fig. 1G-H. Analyzing localization of two over-expressed proteins is always problematic since one cannot discount the possibility that these proteins co-aggregate when overexpressed. That is especially concerning since GFP-GhRabA4C localization in Figure S2 is quite different when it is co-expressed with ER marker mRFP-HDEL. The colocalization of GFP-GhRabA4c with endogenous Golgi and ER markers needs to be shown. Also, it would be good to know the level of overexpression of GFP-GhRabA4c (as compared to endogenous GhRabA4c).

The author's answer: Thanks for your comments. As you mentioned, co-localization of GFP-GhRabA4c with endogenous Golgi and ER markers is the best way to show GhRabA4C localizes at Golgi and ER. Transgenic plants that harbor Golgi and ER markers are needed to perform these experiments. However, as you may know, making transgenic cotton is not easy, taking more than one year to get the T1 plants. Herein, we alternatively used tobacco to detect the localization of GhRabA4C, and showed that both the active and inactive form of GhRabA4C localize at ER, whereas only the active form of GhRabA4C localize at trans-Golgi. Moreover, tobacco leaves have been extensively used for subcellular localization test in plant protein studies. As we used tobacco leaves to transiently express GFP-GhRabA4c, no endogenous GhRabA4c expression in the tobacco leaves. Thanks for your suggestions again.

2. Figure 2A-B. The change in fiber length is modest at best. The only more dramatic difference is at 1 DPA stage. That questions the importance of GhRabA4c for cell elongation.

The author's answer: Thanks you very much. Your comments made us think more deeply about our studies. The fiber length of *GhRabA4c* overexpression and down-regulation lines was about 1.5 mm longer and 1.0 mm shorter than that of the WT in mature fiber, respectively. Thus, *GhRabA4c* plays an important role in fiber cell elongation. In cotton breeding practice, increasing cotton fiber length by 1.0 mm of mature fiber has great improvement in cotton quality breeding program. We have modified Figure 2B and added the relative results of two 35S::*GhRabA4c* transgenic lines. Please see modified Figure 2B and Appendix Figure S4.

3. Figure 3A. Y2H assays is missing some key controls. First, colonies of Double-drop out media needs to be shown to ensure that "negative" control yeast are viable. Second, pGBKT7-GhRabA4c and pGADT7-GhACT4 alone needs to be shown to control for possible auto-activation.

The author's answer: Thanks for your comments. We have added the growth condition of yeast colonies on double-drop out medium. Please see the revised Figure 3A. In addition, we have performed the auto-activation test of pGBKT7-GhRabA4c, pGADT7-GhACT4 and pGADT7-GhACT1. Please see appendix Figure S6 for auto-activation test of GhRabA4c, GhACT4 and GhACT1 proteins. None of GhRabA4c, GhACT4 and GhACT1 protein has auto-activation activity.

4. Figure 3B. Why GFP-GhRabA4c localization is so different compared to the ones shown in colocalization with Golgi experiments?

The author's answer: Thanks for your comments. Different planes in the three-dimensional cell were taken photos when observing co-localization of GFP-GhRabA4c with Golgi or actin filaments. Same as previous studies, we also found that the actin filaments always stretched out from the nucleus. Thus, the co-localization of GFP-GhRabA4c with actin filaments was observed in the cell plane close to the nucleus, whereas Golgi co-localization was mostly observed in the plane of cytoplasm. We speculate that is the reason lead to difference between Figure 3B and Figure 1F-H. Thank you again.

5. Figure 4. Based on bright field images, the area imaged for SE1 and SE2 is quite different. I am not even convinced these are the vesicles. How was density quantified?

The author's answer: Thanks for your comments. SE1 and SE2 are the two *GhRabA4c* down-regulated lines that exhibit shorter fibers at 1 DPA, as shown by the scanning electron microscope in Figure 2C. In Figure 4, the fiber cells at 2 DPA were used, and there is a great possibility that fiber cells of SE1 and SE2 are still shorter than WT and OE1/2. Thus, the bright field images of SE1/2 look different from that of WT and OE1/2. Consistent with the shorter fibers, vesicles in SE1 and SE2 fibers were aggregated into patches, implying the functions of GhRabA4c in vesicle trafficking.

Regarding to the vesicle density quantification, we first measured the total areas of fiber cells using ImageJ software. Then we manually counted the number of stained vesicles in fiber cell.

Vesicle density was expressed as the vesicle number per square millimeter. At least three biological replicates were used in each sample. Thanks again for your questions.

6. Figure 5. Not quite sure about scientific rationale by doing RNAseq analyses of lines expressing different levels of GhRabA4c. If GhRabA4c regulates membrane transport and actin dynamics, why authors would expect that it will directly affect transcription?

The author's answer: Thanks for your questions. We believe that cotton fibers coordinate several extracellular or intracellular metabolic activities to achieve the elongation process. Metabolic activity changes in one pathway, for example, the membrane transport and actin dynamics, will disturb the metabolic balance in the cell, and the cell will adapt to these changes via feedback regulation. One regulation mechanism is to reprogram its transcription. Based on these perspectives, we performed the RNA-seq analysis on WT and GhRabA4c transgenic lines. Interestingly, as expected, pathways related to intracellular lumen and cytoskeleton network were enriched. And genes involved in cell wall polysaccharide biosynthesis were also found to be differentially expressed between WT and GhRabA4c transgenic lines. These RNA-seq analyses provide further evidence for the importance of GhRabA4c in regulating vesicle trafficking and actin dynamics. Thanks again for your comments.

7. Figure 5E-L. How authors know that changes in particular gene expression is the result of GhRabA4c levels or simple clonal variations. Do the same changes in expression observed (by qPCR) if using OE2 or SA2 lines?

The author's answer: Thanks for your comments. Following your suggestion, we performed qPCR analysis on WT, OE1, OE2, SE1 and SE2, and the same changes in expression observed in OE2 or SE2 lines. The results were shown in the modified Figure 5E-L. Please see that. Thanks again.

August 15, 2022

RE: Life Science Alliance Manuscript #LSA-2022-01450-TR

Prof. Wangzhen Guo
Nanjing Agricultural University
Weigang No1
Nanjing, Jiangsu 210095
China

Dear Dr. Guo,

Thank you for submitting your revised manuscript entitled "GhRabA4c coordinates cell elongation via regulating actin filament-dependent vesicle transport". We would be happy to publish your paper in Life Science Alliance pending final revisions necessary to meet our formatting guidelines.

- Please combine Figure S6 and S7 as suggested by Reviewer 1
- Please include the appendix method file into your methods section in the main manuscript file
- please add ORCID ID for corresponding authors-you should have received instructions on how to do so
- please add the Twitter handle of your host institute/organization as well as your own or/and one of the authors in our system
- please add your supplementary figure legends and your table legends to the main manuscript text; please refer to your supplementary figures as supplementary rather than appendix figures
- we encourage you to introduce your figure panels in your figure legends in alphabetical order

Figure Check:

- Scale bars needed for Figure S9D and Figure S10

A. FINAL FILES:

B. MANUSCRIPT ORGANIZATION AND FORMATTING:

Sincerely,

Reviewer #1 (Comments to the Authors (Required)):

In this revised manuscript the authors provide additional data to support the function of GhRabA4c in fiber elongation. Importantly, they show that GhRabA4c interacts with fiber-specific GhACT1, which supports the role of GhRabA4c in fiber elongation and indicates the important role of this interaction. They also showed results of longer fibers of the 35S::GhRabA4c overexpression plants, which are consistent with those of RDL1::GhRabA4c plants. In addition, the possible mechanism of how GhRabA4c coordinates multiple cellular processes to change cell shape and enhance fiber elongation, as well as how the intracellular transportation affects the cell wall structure is discussed.

Minor point:

Appendix Figure S6 and S7 can be combined, as pictures in Fig S6 can be considered as negative controls of GhRabA4c-GhACT1 interaction.

Reviewer #3 (Comments to the Authors (Required)):

For most part reviewers addressed my comments. Thus, the paper is suitable for publication.

Response to Review 1

Thank you for your review. The authors had accepted and corrected the MS as you suggested. The explanations point to point as follows:

Minor point:

Appendix Figure S6 and S7 can be combined, as pictures in Fig S6 can be considered as negative controls of GhRabA4c-GhACT1 interaction.

The author's answer: Thanks for your comments. We have combined these two supplemental figures. Please see the updated Figure S6.

August 19, 2022

RE: Life Science Alliance Manuscript #LSA-2022-01450-TRR

Prof. Wangzhen Guo
Nanjing Agricultural University
Weigang No1
Nanjing, Jiangsu 210095
China

Dear Dr. Guo,

Thank you for submitting your Research Article entitled "GhRabA4c coordinates cell elongation via regulating actin filament-dependent vesicle transport". It is a pleasure to let you know that your manuscript is now accepted for publication in Life Science Alliance. Congratulations on this interesting work.

DISTRIBUTION OF MATERIALS:

Again, congratulations on a very nice paper. I hope you found the review process to be constructive and are pleased with how the manuscript was handled editorially. We look forward to future exciting submissions from your lab.

Sincerely,
